# Pan-cancer characterization of immune-related lncRNAs identifies potential oncogenic biomarkers

Yongsheng Li [1,2,3,4,5✉], Tiantongfei Jiang[1,4], Weiwei Zhou[1,4], Junyi Li[1], Xinhui Li[1], Qi Wang[1], Xiaoyan Jin[1], Jiaqi Yin[1], Liuxin Chen[1], Yunpeng Zhang[1], Juan Xu[1,2,3,5✉] & Xia Li[1,2,3,5✉]

Long noncoding RNAs (lncRNAs) are emerging as critical regulators of gene expression and they play fundamental roles in immune regulation. Here we introduce an integrated algorithm, ImmLnc, for identifying lncRNA regulators of immune-related pathways. We comprehensively chart the landscape of lncRNA regulation in the immunome across 33 cancer types and show that cancers with similar tissue origin are likely to share lncRNA immune regulators. Moreover, the immune-related lncRNAs are likely to show expression perturbation in cancer and are significantly correlated with immune cell infiltration. ImmLnc can help prioritize cancer-related lncRNAs and further identify three molecular subtypes (proliferative, intermediate, and immunological) of non-small cell lung cancer. These subtypes are characterized by differences in mutation burden, immune cell infiltration, expression of immunomodulatory genes, response to chemotherapy, and prognosis. In summary, the ImmLnc pipeline and the resulting data serve as a valuable resource for understanding lncRNA function and to advance identification of immunotherapy targets.

[1] College of Bioinformatics Science and Technology, Harbin Medical University, Harbin, Heilongjiang 150081, China. [2] Key Laboratory of Tropical Translational Medicine of Ministry of Education, Hainan Medical University, Haikou 571199, China. [3] College of Biomedical Information and Engineering, Hainan Medical University, Haikou, Hainan 570100, China. [4]These authors contributed equally: Yongsheng Li, Tiantongfei Jiang, Weiwei Zhou. [5]These authors jointly supervised this work: Yongsheng Li, Juan Xu, Xia Li. ✉email: liyongsheng@ems.hrbmu.edu.cn; xujuanbiocc@ems.hrbmu.edu.cn; lixia@hrbmu.edu.cn

Misregulation of gene expression programs has been found to cause a broad range of human diseases[1]. Gene expression in humans is controlled by thousands of regulators, such as transcription factors, chromatin regulators, and noncoding RNAs (ncRNAs)[2]. Cancer transcriptome analysis has identified thousands of lncRNAs that are associated with different types of cancer[3]. However, knowledge about their function in cancer is still limited.

Much research has been focused on the function of lncRNAs and their mechanisms of action are even more diverse. Early studies primarily demonstrated that lncRNAs are related to diverse cellular responses, such as cell proliferation, apoptosis, and differentiation[4–6]. In addition, increasing studies have reported that the tumor microenvironment plays important roles in cancer development and progression. Dysregulation of the immune system can be a major cause of the development of cancer and immunotherapy has emerged as a promising cancer treatment strategy[7]. Thus, precise regulation of the expression of immune genes is critical for generating a robust immunity. However, the majority of studies so far have focused on coding genes, particularly the function of cell-surface receptors, cytokines, and transcription factors. Recently, increasing evidence has revealed that lncRNAs can play fundamental roles in the regulation of the immune system[8,9]. For example, the lncRNA *NeST* has been found to be related with T-cell activation and to be critical for immune response regulation[10]. The lncRNA *NRON* has been demonstrated to maintain a resting state of T cells by sequestering phosphorylated *NFAT* in the cytoplasm[11]. The oncogenic lncRNA *LINK-A* downregulates cancer cell antigen presentation and intrinsic tumor suppression[12]. However, only a few immune-related lncRNAs have been found to play a role in cancer so far[12–14]. Therefore, further studies on lncRNAs and their roles in immune regulation will be essential to identify immunotherapy targets in cancer.

To systematically identify immune-related lncRNAs that are involved in cancer, we integrated multi-omics data of 33 cancer types and proposed the ImmLnc pipeline. We identified several lncRNAs that are related to immune pathways, which we further validated by independent datasets. We found that these immune lncRNAs are likely to be highly expressed in immune cell populations, show expression perturbation in cancer, and are significantly correlated with immune cell infiltration. With the examples of cancer-related lncRNA prioritization and cancer subtyping, we demonstrate that ImmLnc is a valuable resource for investigating the function of lncRNAs in cancer.

## Results

### Identification of immune-related lncRNAs across cancer types.
To identify candidate lncRNA regulators that are correlated with immune-related pathways, we proposed a three-step computational framework called ImmLnc (Fig. 1a). ImmLnc systematically infers candidate lncRNA modulators of immune-related pathway activity from a large collection of sample-matched gene and lncRNA expression profiles. We reasoned that if the lncRNAs play important roles in immunology, then their correlated genes should be enriched in the immune-related pathways. Briefly, ImmLnc identifies the lncRNA modulators in three steps (Fig. 1a). First, genome-wide gene and lncRNA expression profiles of the same tumor patients were collected. Second, we calculated the tumor purity for each patient and genes were ranked based on the *RS* score for each candidate lncRNA. Third, we computed for each lncRNA its activity in immune pathways (lncRES) based on the modified gene set enrichment analysis (GSEA)[15,16]. The *P*-value of GSEA was converted to a lncRES score and the lncRNA–pathway

pairs with lncRES > 0.995 and a false discovery rate (FDR) < 0.05 were selected.

Taking advantage of the multi-omics data in The Cancer Genome Atlas (TCGA), we employed ImmLnc for the genome-wide identification of lncRNA modulators in >11,000 samples across 33 cancer types (Fig. 1b and Supplementary Data 1). In particular, we focused on 17 immunologically relevant gene sets representing distinct immune pathways derived from ImmPort[17], which is one of the largest open repositories of immunological data. Three to 516 genes were involved in these pathways (Fig. 1c and Supplementary Data 2). When ImmLnc was applied to 33 cancer types, the genome-wide screening yielded a collection of lncRNA pathways. On average, each cancer type yielded ~2000 lncRNAs that were correlated with immune pathways (Fig. 1d). Higher number of immune-related lncRNAs were identified in the cancer types in which more lncRNAs are expressed (Supplementary Fig. 1). These immune-related lncRNAs accounted for ~25% of all lncRNAs. Higher number of lncRNAs were correlated with the "Cytokines" and "Cytokine Receptors" pathways across cancer types (Supplementary Fig. 2). The cytokine and chemokine system represents an emerging potential target for immunotherapy[18], and these lncRNA modulators will be a resource for dissecting the immune regulation underlying cancer. Moreover, we applied the ImmLnc pipeline to two datasets across immune cell populations (details are described in the Methods section). We found that the lncRNA–pathway pairs significantly overlapped with those identified in tissue datasets (Supplementary Data 3). All these results suggest that ImmLnc is able to identify the critical lncRNAs involved in immunology and constitutes a valuable resource in the development of precision medicine.

### Expression perturbation of lncRNA regulators in cancer.
To gain insight into the function of immune-related lncRNAs, we further examined the 500 top-ranked lncRNA–pathway pairs that are identified in multiple cancer types (Fig. 2a and Supplementary Data 4) and identified a lncRNA–pathway regulatory network. This regulatory network involves 241 lncRNAs and 13 immune pathways. The majority of these lncRNAs were correlated with "Cytokines" (Supplementary Fig. 3a). In addition, we investigated the expression of these lncRNAs and found that a large number of lncRNAs are upregulated in cancer (Fig. 2a). For example, *MIAT*, which has been demonstrated to be an oncogenic lncRNA by promoting the cell growth and metastasis[19], shows more than threefold upregulation in ten cancer types (Supplementary Fig. 3b). *PVT1*, which has been reported to play vital roles in a variety of cancers[20], shows expression perturbation in 15 cancer types (Supplementary Fig. 3c). To further investigate which specific cytokine pathways were correlated with these two lncRNAs, we obtained the "IFNG_score" and "Cytotoxic cells" score of each patient[21]. We found that the expression of *MIAT* and *PVT1* was significantly correlated with the pathway scores (Supplementary Fig. 4). Moreover, we found that these cytokine-related lncRNAs are likely to co-occur with "cytokine" in the literature (Supplementary Fig. 5). These results imply that they might play critical roles in cytokine-related pathways.

To preliminarily validate the results obtained with ImmLnc, we first explored whether the lncRNA–pathway pairs could be reproduced with different datasets of the same cancer type. We collected another six sample-matched gene and lncRNA expression profiles from The International Cancer Genome Consortium[22] and Gene Expression Omnibus. The lncRNA pathways were again identified by ImmLnc in each cancer type separately. We found significant lncRNA pathway overlap for the same cancer type (Fig. 2b, *P* < 0.001 for all cancer types, two-sided

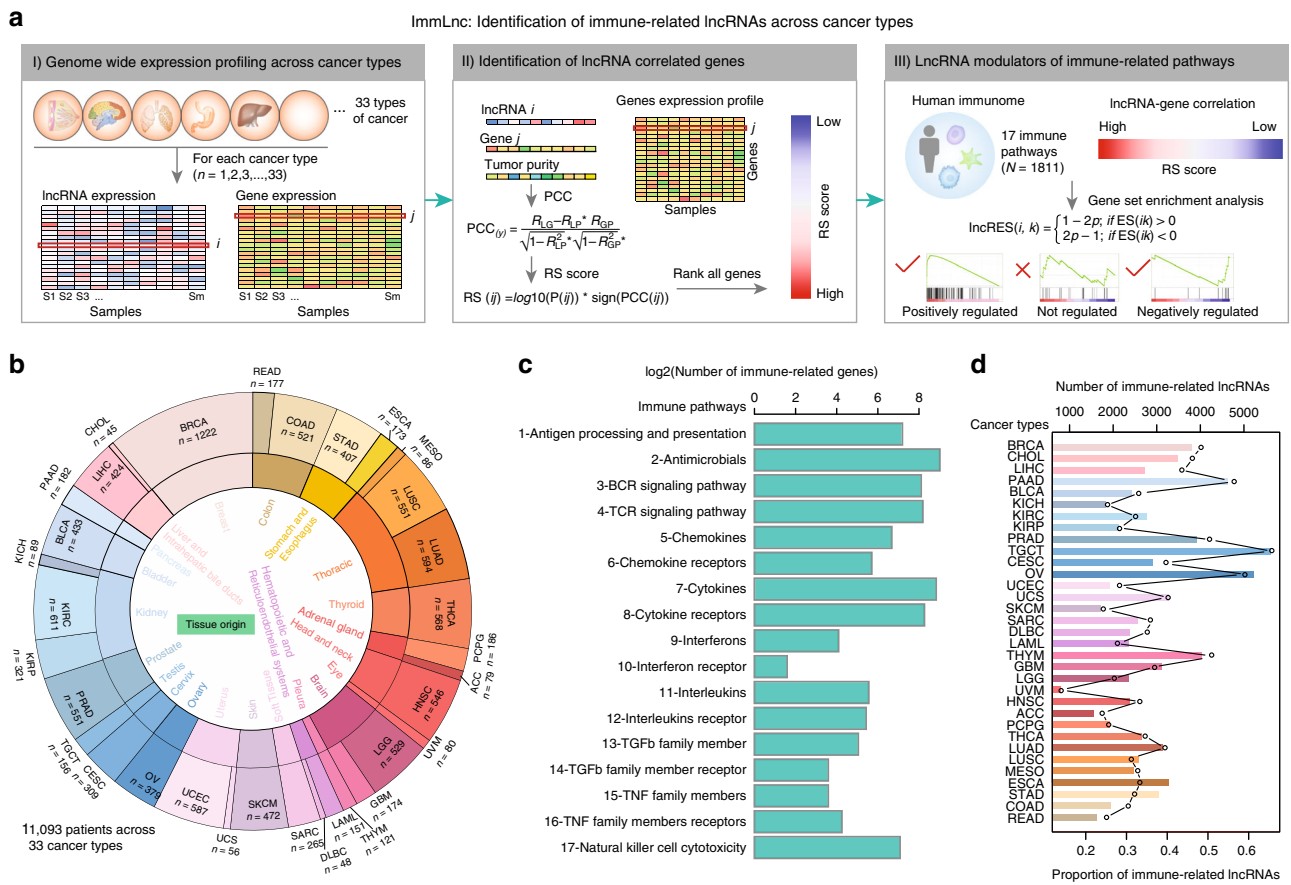

**Fig. 1 Identification of immune-related lncRNAs across cancer types. a** Schematic illustration of three steps of ImmLnc for the identification of lncRNA regulators. **b** The number of patients in each cancer type. Cancers with similar tissue origin are grouped together. **c** The number of genes in 17 immune-related pathways. **d** The number of immune-related lncRNAs identified in each cancer type. The top *y*-axis shows the number of lncRNAs and the bottom *y*-axis shows the proportion of lncRNAs. Source data are provided as a Source Data file.

hypergeometric test). These results suggest that the algorithm was effective in reproducing the immune-related lncRNAs.

In addition, to further validate ImmLnc, we explored whether these lncRNA modulators play important roles in cancer. Thus, we first identified all the lncRNAs that show expression perturbations in 17 cancer types, which are with more than five normal samples. Interestingly, the immune-related lncRNAs were more likely to show expression perturbation than other lncRNAs across cancer types (Fig. 2c, all *P* < 0.001, two-sided Fisher's exact tests), particularly in the cancer types suitable for immunotherapy (such as kidney and lung cancer)[23,24]. For example, ~30% of immune lncRNAs exhibited expression perturbation in kidney renal clear cell carcinoma; this percentage is more than twice as high as that of the total lncRNAs. Moreover, we found that cancer types with similar tissue origin significantly share immune-related lncRNAs (Supplementary Fig. 3d), such as low-grade glioma and glioblastoma multiforme, and colon adenocarcinoma and rectum adenocarcinoma. Such molecular events can be utilized as surrogate biomarkers for early detection. Collectively, these results suggest that ImmLnc is a useful pipeline to identify immune regulators that exhibit perturbed expression.

**LncRNA are associated with immune cell infiltration.** The immune reaction has been demonstrated to involve tissue infiltration of immune cells[25]. Therefore, we reasoned that if lncRNAs identified by ImmLnc play important roles in immune regulation, then they would be more likely to be highly expressed in immune

cells and to be correlated with immune cell infiltration in tumors. We first analyzed ten single-cell sequencing data for immune cells downloaded from PanglaoDB[26]. A significantly higher proportion of immune-related lncRNAs was expressed in immune cells (Supplementary Fig. 6). Moreover, we derived the lncRNA expression profile in immune cells of TCGA bulk RNA sequencing (RNA-Seq) samples based on the ideas from RESPECTEx[27] (see details in Supplementary Methods). We also found that these lncRNAs were significantly more highly expressed in immune cells (Supplementary Fig. 7). In particular, we analyzed another single-cell sequencing data for lung cancer[28]. We found that immune-related lncRNAs identified in lung cancer exhibited significantly higher expression in B cells and T cells of lung cancer patients (Supplementary Fig. 8). These results suggest that immune-related lncRNAs exhibit higher expression in immune cell populations.

We next estimated the immune cell infiltration levels of each patient based on gene expression by Tumor Immune Estimation Resource (TIMER)[29]. Here, six tumor-infiltrating immune cells (B cells, CD4 T cells, CD8 T cells, macrophages, neutrophils, and dendritic cells) were considered. The associations between immune infiltrates and lncRNA expression were evaluated by Spearman's rank correlation coefficient (|*R*| > 0.3 and *P* < 0.05). We found that a large number of lncRNAs that were correlated with immune cell infiltration were immunology related (Fig. 3a and Supplementary Data 5). LncRNAs that were correlated with immune cell infiltration also exhibited higher expression in immune cells (Supplementary Figs. 9 and 10). For example, of the

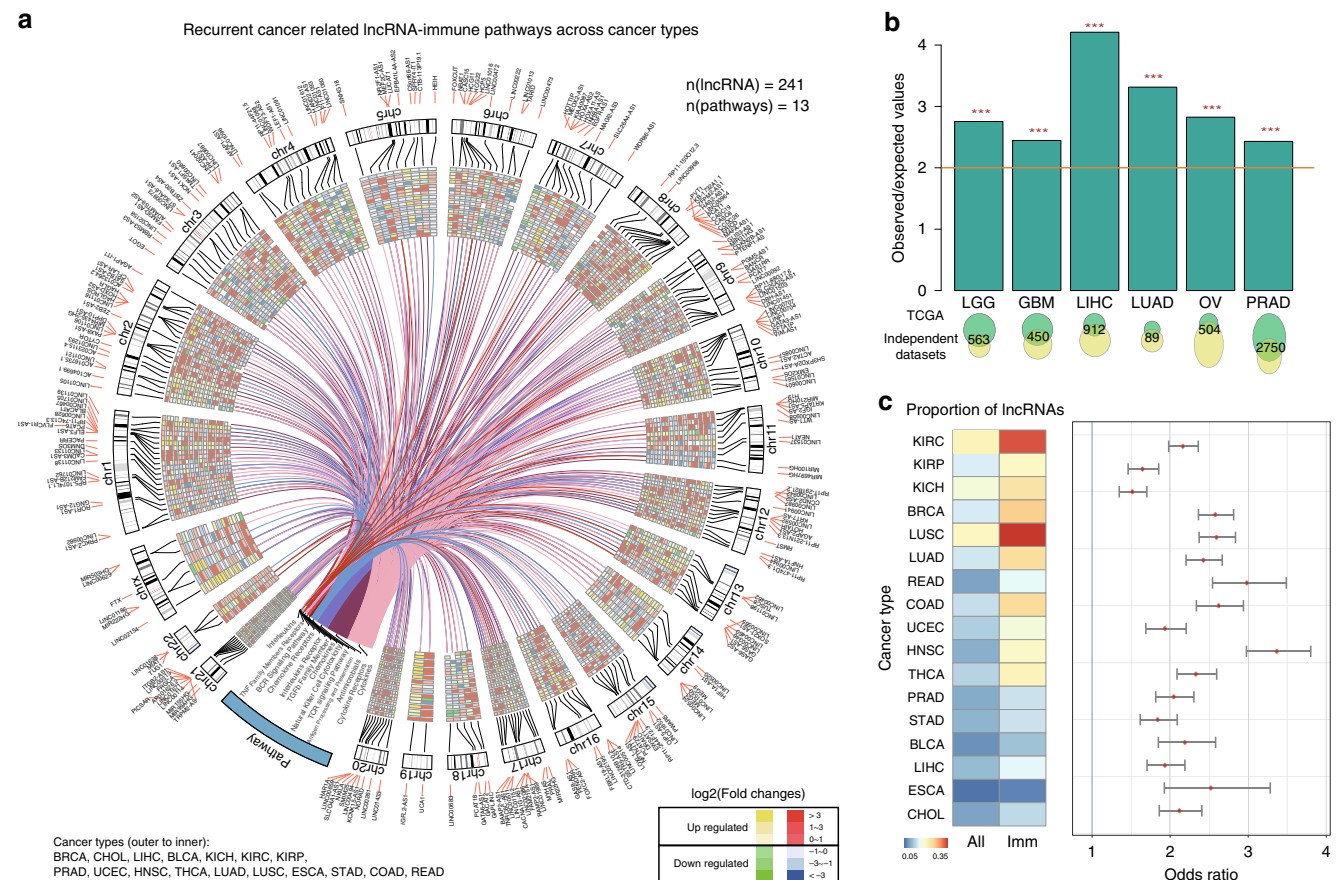

**Fig. 2 Validation of immune-related lncRNAs. a**. Circos plot showing the 500 top-ranked lncRNA–pathway pairs across cancer types. The inner heat map shows the differential expression of lncRNAs across cancer types. Red and yellow colors represent upregulation; blue and green colors represent downregulation. **b** The overlap of immune–lncRNA pathways in independent datasets of the same cancer type. The top bar plots show the observed/expected values of the hypergeometric test. The bottom Venn plots show the number of overlapping lncRNA–pathway pairs. ***all $P < 2.2E - 16$, two-sided hypergeometric tests. **c** The left heat map shows the proportion of differentially expressed lncRNAs of all lncRNAs and of immune-related lncRNAs. The right panel shows the odds ratios in each cancer type. All $P < 0.01$, two-sided Fisher's exact tests. The error bars were 95% confidence levels of odds ratios. Source data are provided as a Source Data file.

lncRNAs whose expression was correlated with immunology in skin cutaneous melanoma (SKCM), 55.62% were related with CD8 T-cell infiltration. The proportion of lncRNAs that are correlated with CD8 T-cell infiltration is much higher in two types of lung cancer (Fig. 3a, 79.15% for lung adenocarcinoma (LUAD) and 68.50% for lung squamous cell carcinoma (LUSC)). In particular, we found that the cancer-related lncRNAs *PVT1* and *MIAT* were significantly correlated with CD8 T-cell infiltration in several cancer types (Supplementary Fig. 11). These results suggest that lncRNAs might play critical roles in immunology regulation.

Moreover, we used Fisher's exact test to investigate whether the lncRNAs identified by ImmLnc were likely to be associated with immune cell infiltration. We found that in the majority of cancer types, a significantly higher proportion of immune-related lncRNAs are correlated with immune cell infiltration (Fig. 3b±g).

Recent studies suggested that genes, whose expression is negatively correlated with tumor purity and positively correlated with immune cell infiltration, are likely to play important roles in immunology[27,30]. We found that these lncRNAs showed a significant overlap with immune-related lncRNAs across cancer types (Supplementary Fig. 12). In particular, there are 84.38% and 56.25% cancer types with significant difference in CD4 and CD8 T-cell infiltration between the immune-related and all lncRNAs. In addition, we found that the immune-related lncRNAs are more

likely to be associated with CD8 T-cell infiltration in melanoma compared with others (Fig. 3d, OR = 4.52, P = 2.51E − 57, two-sided Fisher's exact test). CD8 T-cell infiltration has been demonstrated to be a useful biomarker for prediction prognosis and response to therapy[31,32]. The identification of these immune-related lncRNAs provides insight into molecular mechanisms regulating T-cell infiltration and activity in tumors. Therefore, we next focused on 70 lncRNA regulators that are significantly correlated with CD8 T-cell infiltration in melanoma (R > 0.6 and P < 0.05; Supplementary Fig. 13a). Several lncRNAs have been demonstrated to play crucial roles in immune regulation. For example, we found that the expression of HLA class I histocompatibility antigen protein P5 (HCP5) is significantly correlated with CD8 T-cell infiltration in SKCM (Supplementary Fig. 13b, R = 0.603, P < 2.2E − 16). A previous study has shown that *HCP5* is expressed primarily in immune system organs, such as the spleen and thymus[33]. These results suggest *HCP5* plays a role in immune regulation. Another prominent example is the lncRNA *ITGB2-AS1*, which is significantly correlated with CD8 T-cell infiltration (Supplementary Fig. 13b, R = 0.66, P < 2.2E − 16). *ITGB2-AS1* has been shown to promote migration and invasion in cancer[34], and is known to be involved in the regulation of T-cell and B-cell activation[35]. Moreover, we identified the lncRNAs that were correlated with immune cells by CIBERSORT[36]. We found that they significantly overlapped

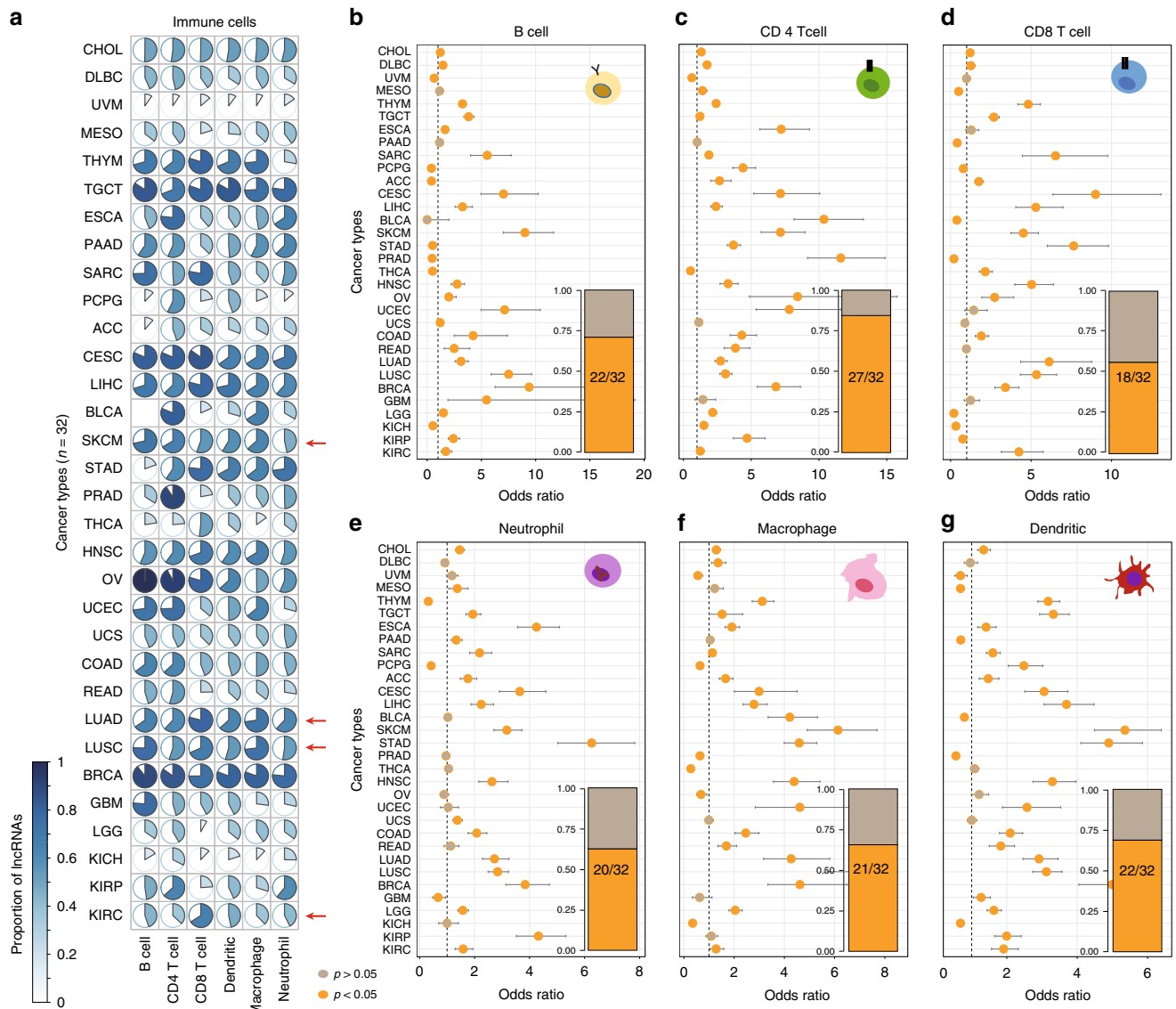

**Fig. 3 The immune lncRNAs are correlated with immune cell infiltration in cancer. a** The proportion of immune lncRNAs that were correlated with immune cell infiltration. **b–g** The immune-related lncRNAs were likely to be enriched in lncRNAs that were correlated with immune cell infiltration. The dots represent the odds ratio (OR) of Fisher's exact test and the error bars show the 95% confidence intervals of the OR. The numbers inside bar plots show the proportion of cancers, where orange indicates cancers with *P*-values < 0.05 and OR > 1 in two-sided Fisher's exact test. **b** B cells; **c** CD4 T cells; **d** CD8 T cells; **e** neutrophils; **f** macrophages; and **g** dendritic cells. Source data are provided as a Source Data file.

with TIMER (Supplementary Data 6). Collectively, all these results suggest that these immunology-related lncRNA regulators exhibit higher expression in immune cells. They are likely to be associated with immune cell infiltration, further validating the results of the ImmLnc algorithm.

**ImmLnc prioritizes cancer-related lncRNAs.** Having demonstrated that ImmLnc can effectively identify immune-related lncRNAs, we next explored the application of ImmLnc for prioritizing cancer-related lncRNAs. Although an increasing number of lncRNAs have been identified by experimental and computational methods[3,37], the identification of novel cancer lncRNAs based on large-scale experiments is expensive and time consuming. We hypothesized that if the lncRNA–pathway relationships can be observed in more cancer types, they are more likely to be involved in cancer. To assess the ability of ImmLnc to prioritize cancer-related lncRNAs, we ranked each

lncRNA–pathway based on the number of cancer types in which they were observed. We found that >75% of the immune-associated lncRNAs were antisense and intergenic (Supplementary Fig. 14). Thus, we focused on 5050 long intergenic ncRNAs (lincRNAs) and each immune-related pathway was analyzed separately. In addition, we ranked lncRNAs based on the number of cancers that showed transcriptome perturbations. The ranks of lncRNAs were normalized (assigned a value between 0 and 1), and we further calculated the average rank for each lncRNA across 17 immune pathways and expression perturbation. We found that the known cancer-/disease-related lincRNAs have significantly higher ranks than other lincRNAs (Fig. 4b and Supplementary Fig. 15a; *P* < 0.001, one-sided Wilcoxon's rank-sum test). In addition, we found that integration of this immune regulation information has a higher area under the receiver-operating characteristic curve than a differential expression-based method (Supplementary Fig. 15b–c). We obtained similar results

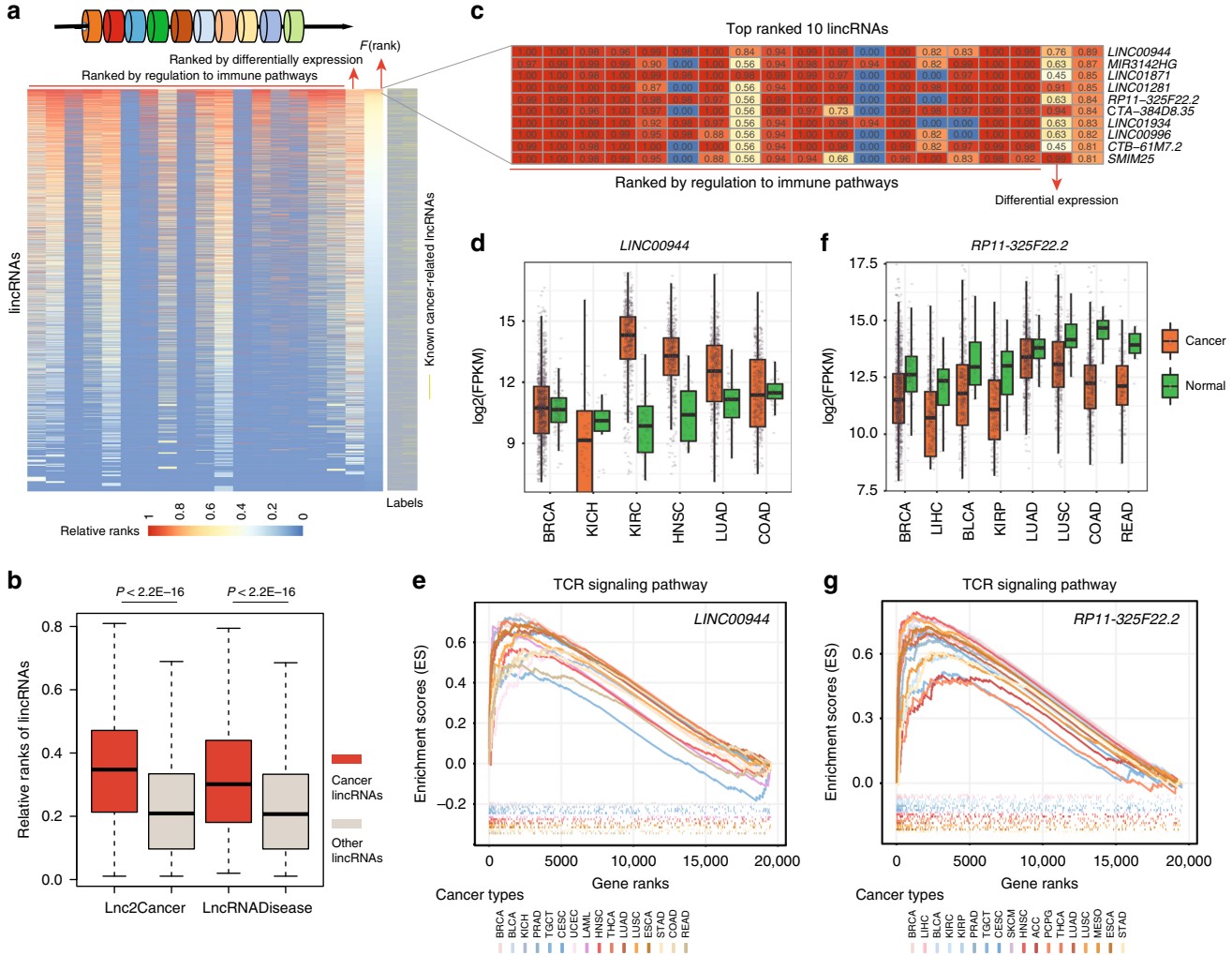

**Fig. 4 Prioritization of cancer-related lncRNAs based on immune regulation. a** The distribution of average rank for lincRNAs. Each column is the normalized rank score for lincRNAs. The first 17 columns are based on immune pathway regulation and the 18th column is based on differential expression. The last column is the final score for each lincRNA. The known cancer-related lncRNAs are colored in yellow. **b** The relative rank score distribution of cancer-related lincRNAs and other lincRNAs. ***$P < 2.2E - 16$, one-sided Wilcoxon's rank-sum test. **c** Heat map showing the normalized rank scores for the ten top-ranked lincRNAs. **d** The distribution of expression for *LINC00944* in different cancer types. **e** The enrichment score (ES) distribution for the LINC00944–TCR (T cell receptor) signaling pathway. Each vertical bar represents a gene involved in the pathway. Each line is for one cancer type. **f** The distribution of expression for RP11-325F22.2 in different cancer types. **g** The enrichment score (ES) distribution for RP11-325F22.2–TCR signaling pathway. Each vertical bar represents a gene involved in the pathway. Each line is for one cancer type. The center of the boxplots are median values, the bounds of the boxes are 25% and 75% quantiles. The minima are 25% quantile − 1.5*interquartile range (IQR) and the maxima are 75% quantile + 1.5*IQR. Source data are provided as a Source Data file.

when we considered all immune-related lncRNAs (Supplementary Fig. 16). These results suggest that ImmLnc can prioritize cancer-/disease-related lncRNAs.

Based on the relative ranks of lincRNAs, we identified several cancer-related lincRNAs, such as *LINC00944* and *SMIM25*[38,39] (Fig. 4c). *LINC00944* has been demonstrated to participate in the process of liver metastasis in colorectal cancer[39]. We found that this lincRNA shows a significant expression perturbation in six cancer types, including breast, kidney, lung, and colorectal cancer (Fig. 4d). Based on the ImmLnc method, we revealed that it is associated with the TCR signaling pathway in 16 cancer types (Fig. 4e). Moreover, we identified several novel candidates that may play critical roles in cancer by regulating immune-related pathways (Fig. 4c). For example, we ranked *RP11-325F22.2* as the fifth lincRNA and we found that this lincRNA shows consistently lower expression in eight cancer types (Fig. 4f). In addition, ImmLnc found this lincRNA was associated with the TCR

signaling pathway in 18 cancer types (Supplementary Fig. 17). These results suggest that *RP11-325F22.2* might be a novel cancer-related lincRNA. Identification of the differentially expressed lncRNAs in cancer is a commonly used method to discover drivers in cancer[40]. However, we found that integration of the lncRNA–immune pathway regulation can prioritize several lincRNAs that cannot be identified by expression perturbation analysis (such as *MIR3142HG* and *CTB-61M7.2*; Fig. 4c). In summary, these results suggest that integration of the ImmLnc results can help prioritizing cancer-related lincRNAs and provide a pathway-based view to improve our understanding of their regulatory function in cancer.

**Immune lncRNAs classify molecular subtypes of cancer.** In addition to identification of the critical molecules in cancer, cancer subtyping is key to the improvement of personalized treatment[41]. Therefore, we next investigated to what extent the

lncRNAs identified by ImmLnc can be applied to molecular cancer subtyping. Lung cancer remains the leading cause of death from cancer around the world[42]. We first identified the lncRNAs that can regulate immune-related pathways in two types of lung cancer (LUAD and LUSC). These lncRNAs overlapped with those that are correlated with infiltration of six immune cell types. We identified 28 common lncRNAs in pan-lung cancer, which were more highly expressed in B and T cells (Supplementary Fig. 18). Next, we found that based on the expression of these 28 lncRNAs, we can classify lung cancer patients into three subtypes (Fig. 5a and Supplementary Fig. 19a). The majority of these lncRNAs are highly expressed in C2 and C3 patients (Supplementary Fig. 19b). Although there are more male patients of the C1 subtype (Fig. 5b, $P = 0.002$, two-sided Fisher's exact test), no difference in cancer stages and smoking status was observed between the three subtypes (Supplementary Fig. 19c–d).

Next, we compared the molecular characterization of these three subtypes. We found C1 patients exhibited the highest tumor differentiation (Fig. 5c; $P < 0.001$, two-sided Wilcoxon's rank-sum test). C2 patients also exhibited higher differentiation scores than C3 patients (Fig. 5c; $P < 0.001$, two-sided Wilcoxon's rank-sum test). Moreover, C1 patients exhibited significantly higher hippo pathway activity, cell cycle pathway activity, and stemness scores (Fig. 5d and Supplementary Fig. 19e). No difference in hippo pathway activity was observed between C2 and C3, but the cell cycle pathway activity in C2 patients was significantly higher than in C3 patients (Fig. 5d; $P < 0.001$, two-sided Wilcoxon's rank-sum test). These results suggest a higher cell proliferation rate in C1 patients. Upon comparison of the survival rates among the three subtypes, we found that C1 patients had poorer prognosis than other patients (Fig. 5e, log-rank test $P = 0.02$). The difference between C1 and C2 was large (log-rank test $P = 0.009$), and after 4000 days the number of alive C2 patients was approximately twice that of alive C1 patients. We next explored the mutations among these three subtypes. We found that several well-known cancer genes exhibit different mutation frequencies among subtypes (Supplementary Fig. 19f), including *TP53*, *KRAS*, *CDKN2A*, and *B2M*. Upon investigation of the expression of genes in the hippo and cell cycle pathways, we found that these genes show higher expression in C1 patients (Supplementary Fig. 19g). Moreover, we compared the lncRNA-based subtypes with other published ones[43] and found that C1 patients are likely to exhibit higher levels of be enriched in DNA copy number-based S1 subtype, protein-based S6 subtype, and AD1 subtype (Fig. 5f). These results collectively suggest that C1 patients have higher cell proliferation and differentiation rates.

**Immune profiling suggests existence of immunology subtype.** The tumor mutation burden (TMB) is emerging as a potential biomarker for immunotherapy[44,45]. Upon examination of the TMB distribution across lung cancer patients, we found that the C2 patients exhibit higher TMB than patients of other subtypes (Fig. 6a and Supplementary Fig. 20a). Moreover, homologous recombination deficiency (HRD) remains an important biomarker and a potentially effective adjunct to enhance immunogenicity of tumors[46]. We compared the HRD scores across different subtypes. We found that C1 and C2 patients have significantly higher HDR scores than C3 patients (Fig. 6b, $P < 0.001$, one-sided Wilcoxon's rank-sum test). Moreover, we explored 160 immune-related gene signatures across patients and revealed clear differences between patients of different subtypes (Fig. 6c, d). Of these gene signatures, 86.88% (139/160) show significant differences among cancer subtypes (Fig. 6d, $P < 0.05$, analysis of variance test). Specifically, we found that C2 and C3 patients had significantly higher B-cell and T-cell scores than C1 patients

(Supplementary Fig. 20b, $P < 0.001$, two-sided Wilcoxon's rank-sum test). Moreover, we found much higher levels of immune cell infiltration (such as CD4 and CD8 T cells) in C2 and C3 patients (Supplementary Fig. 20c). The T-cell infiltration levels in C2 patients were significantly higher than those in C3 patients. These results suggest that C2 patients are more likely to respond to immunotherapy.

We next investigated the distributions of the immune, immune cytolytic activity (CYT) and major histocompatibility complex (MHC) scores among lung cancer patients. All these scores have been demonstrated to be useful biomarkers for predicting the immune response[47,48]. We found that C2 patients had significantly higher immune, CYT, and MHC scores than others (Fig. 6e, $P < 0.001$; Kolmogorov–Smirnov test). To examine whether these C2 patients are likely to respond to therapy, we obtained the chemotherapy information of all lung cancer patients from the TCGA project. We did indeed find that a higher proportion of C2 patients responded to chemotherapy (Fig. 6f, $P < 0.001$, two-sided Fisher's exact test). Moreover, we found that C2 patients who received chemotherapy had a significantly better prognosis than C1 patients who received chemotherapy (log-rank test $P = 0.0002$).

Moreover, we examined the gene expression profile of the T-cell signaling pathway. We found that several genes representing potential targets for immunotherapy exhibit higher expression in C2 patients than in C1 patients (Fig. 6g), including *PDCD1* (*PD1*), *CD274* (*PDL1*), *PDCD1LG2* (*PDL2*), and *CTLA4*. The expression of lncRNAs was correlated with several of these genes (Supplementary Fig. 20d). These results suggest that ImmLnc identifies different cancer subtypes (such as the C1 (proliferative) and C2 (immunological) subtypes) with remarkable molecular and immunology diversity (Fig. 6h), which is helpful to improve personalized cancer management.

**ImmLnc: a web-based resource for immune lncRNAs in cancer.** To help researchers apply the principles described in this work to any disease of interest, we have developed a comprehensive and interactive ImmLnc web resource (http://bio-bigdata.hrbmu.edu.cn/ImmLnc). With this platform, users can query the lncRNA or immune-related pathways of interest in specific cancer context (Fig. 7a). It also provides the correlation between the expression of lncRNAs and immune cell infiltration in cancer (Fig. 7b). Moreover, the users can easily investigate whether the lncRNA of interest shows expression perturbation in cancer (Fig. 7c). We have also provided the R package for investigating the lncRNA–pathway correlation based on the paired lncRNA and gene expression across patients (Fig. 7d). All the data generated in this work can be downloaded for further analysis (Fig. 7e) and the query results are shown in a user-friendly way. The features provided in the resource, which will be continuously updated, should serve as a guide for biologists interested in identifying the regulators of immune-related pathways.

**Discussion**
Accumulating evidence suggests that lncRNAs are important for immune regulators. However, only a few examples have been identified so far. In this study, we report the use of the ImmLnc algorithm to systematically identify the lncRNA regulators that potentially regulate immune-related pathways. We have demonstrated that lncRNAs are active participants in immune regulation in 33 cancer types. Notably, the immune-related lncRNAs are likely to exhibit higher expression in immune cells, show expression perturbation, and are significantly correlated with immune cell infiltration. Moreover, by means of two applications we have demonstrated that ImmLnc helps prioritizing cancer-

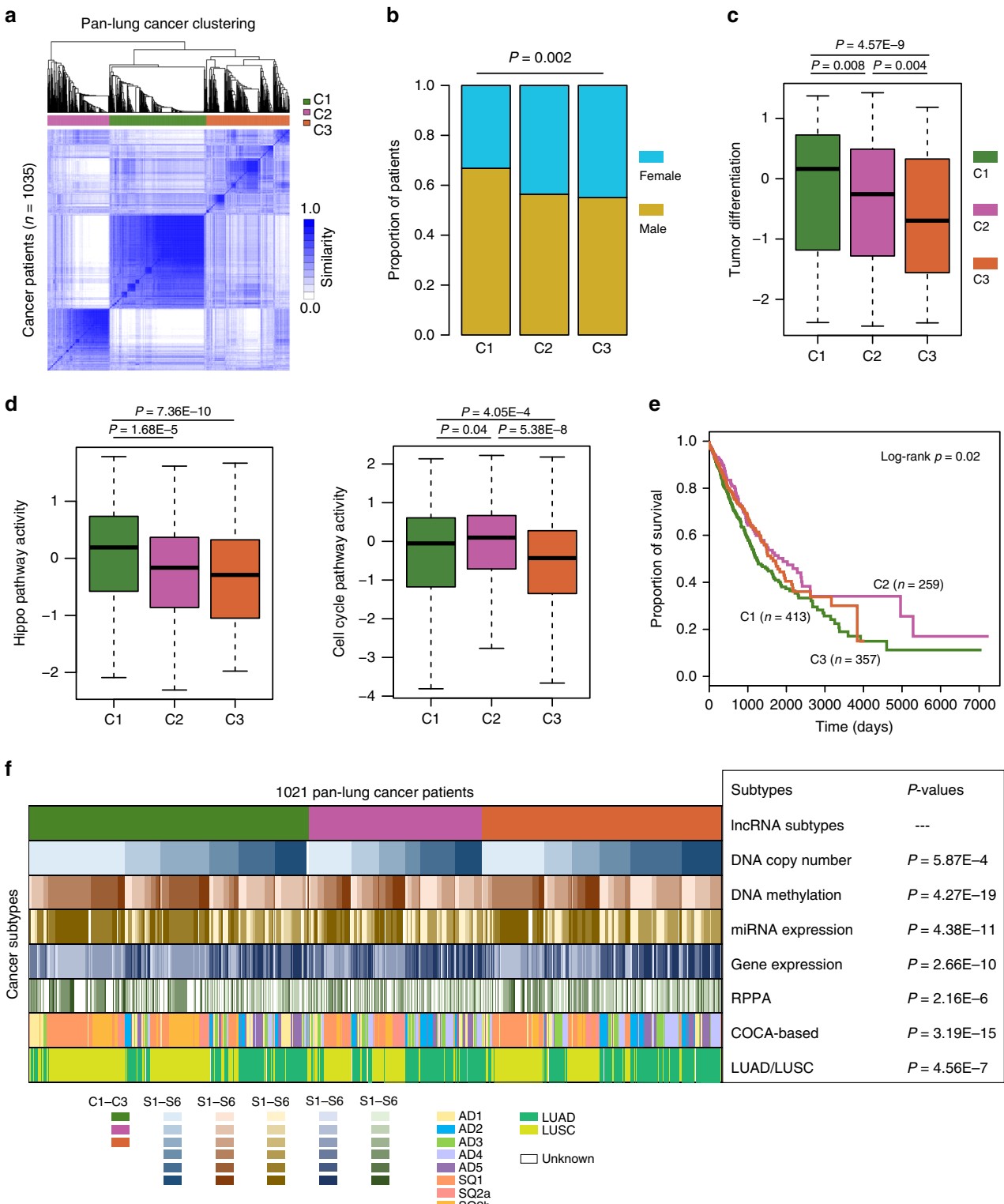

**Fig. 5 Characterization of lung cancer subtypes. a** Heat map of the matrix of co-occurrence proportions for cancer patients. **b** The proportion of male and female patients of different subtypes. n1 = 410, n2 = 257, and n3 = 356 biologically independent samples examined. P = 0.002, two-sided Fisher's exact test. **c** The distribution of tumor differentiation scores for patients of different subtypes. n1 = 411, n2 = 253, and n3 = 357 biologically independent samples examined. P-values for two-sided Wilcoxon's rank-sum tests. **d** The distribution of hippo and cell cycle pathway activity for patients of different subtypes. n1 = 411, n2 = 253, and n3 = 357 biologically independent samples examined. P-values for two-sided Wilcoxon's rank-sum tests. **e** The survival analysis using classifications generated from consensus clustering. C1 patients display a significantly poorer prognosis than C2 and C3 patients. **f**. Comparative analysis of different lung cancer subtypes. The center of the boxplots are median values, the bounds of the boxes are 25% and 75% quantiles. The minima are 25% quantile – 1.5*interquartile range (IQR) and the maxima are 75% quantile + 1.5*IQR. Source data are provided as a Source Data file.

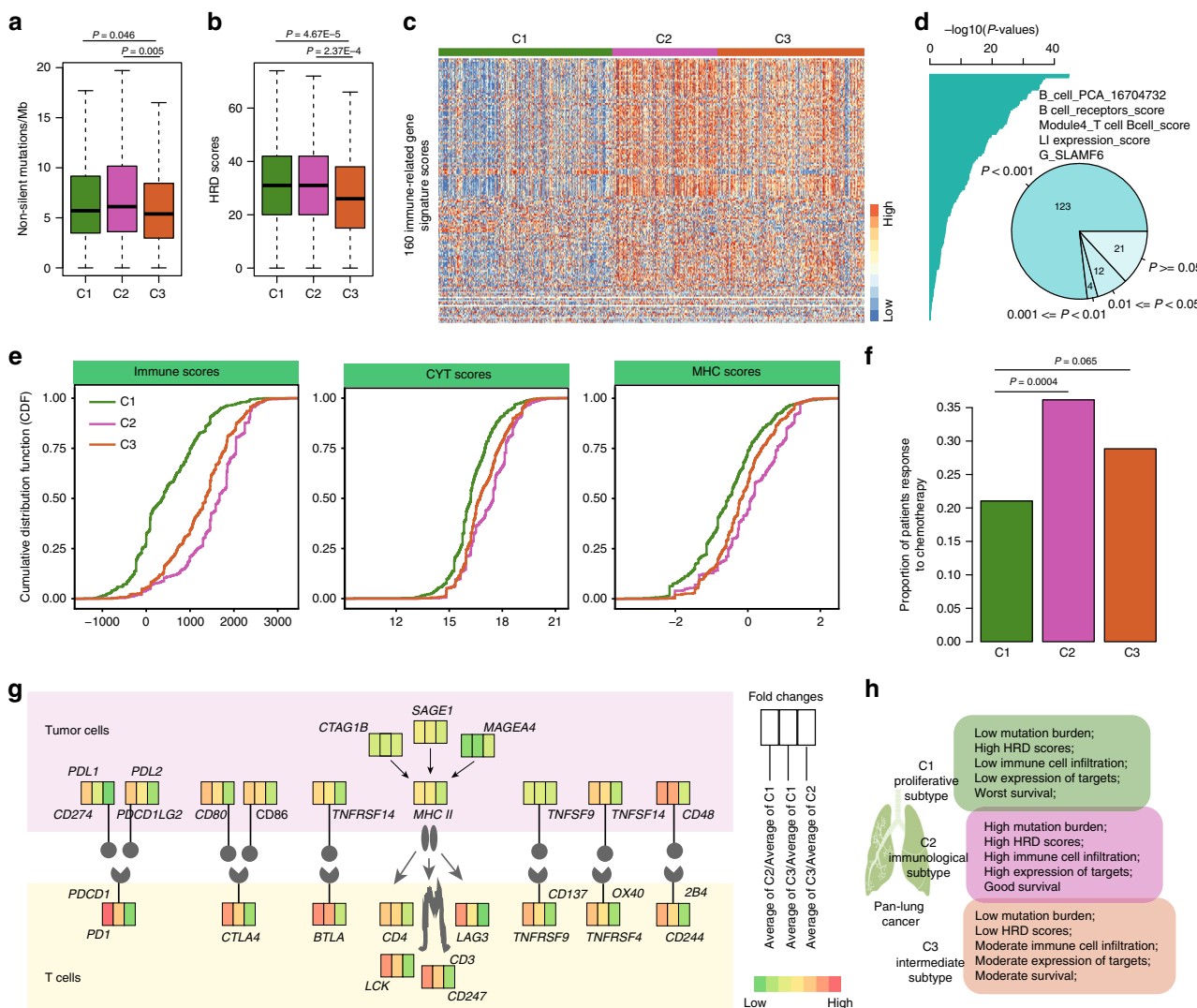

**Fig. 6 The immunological characterization of different lung cancer subtypes. a** The mutation burden for patients in different subtypes. $n1 = 392$, $n2 = 253$, and $n3 = 343$ biologically independent samples examined. *P*-values for one-sided Wilcoxon's rank-sum tests. **b** The HRD score distribution for patients in different subtypes. $n1 = 405$, $n2 = 253$, and $n3 = 352$ biologically independent samples examined. *P*-values for one-sided Wilcoxon's rank-sum tests. **c** Heat map showing 160 immune-related gene signature scores for patients across different subtypes. **d** The distribution of *P*-values obtained by ANOVA analysis. The pie plot shows the proportion of signatures with different *P*-values. **e** The cumulative distribution of immune, CYT, and MHC scores for patients of different cancer subtypes. **f** The proportion of patients that respond to chemotherapy in different subtypes. Two-sided Fisher's exact test. **g** Diagram of the immune checkpoint pathway, comparing tumors in different subtypes. **h** Summary of features for patients of different subtypes. The center of the boxplots are median values, the bounds of the boxes are 25% and 75% quantiles. The minima are 25% quantile − 1.5*interquartile range (IQR) and the maxima are 75% quantile + 1.5*IQR. Source data are provided as a Source Data file.

related lncRNAs and identified cancer subtypes with distinct immunological characterization. The web-based ImmLnc platform provides a valuable resource to investigate the function of lncRNAs in immune regulation.

LncRNAs are emerging as critical regulators of gene expression in the immune system and play critical roles in the development and progression of cancer. However, only a few immune-related lncRNAs have so far been shown to play a role in cancer. Therefore, it is difficult to determine the accuracy of the ImmLnc pipeline directly, so we validated this method by expression perturbation and immune cell infiltration datasets, and showing applications in prioritizing cancer-related lncRNAs and cancer subtyping. Moreover, several lncRNAs associated with cell growth were identified by CRISPR screening, so we indirectly validated the pipeline in identifying essential lncRNAs. We found that the essential lncRNAs were of relatively low rank in cancer

(Supplementary Fig. 21). These results suggest that the pipeline can accurately identify the regulators of a specific functional pathway. Moreover, we found that a significantly higher proportion of immune-related lncRNAs co-occurred with "immune" in the literature (Supplementary Fig. 22). Although we used bulk RNA-Seq data in the current study, we found that the lncRNA–pathway associations significantly overlapped with those identified based on expression in immune cell populations (Supplementary Data 3). With the continuous increase in single-cell sequencing data in immune cells, ImmLnc can help in identifying more lncRNA regulators in immunology.

An increasing number of studies suggest that biomarkers always show tissue-specific expression. Therefore, we analyzed the expression of the immune-related lncRNAs in public sequencing data on various immune cells. We found that the immune-related lncRNAs and the lncRNAs associated with

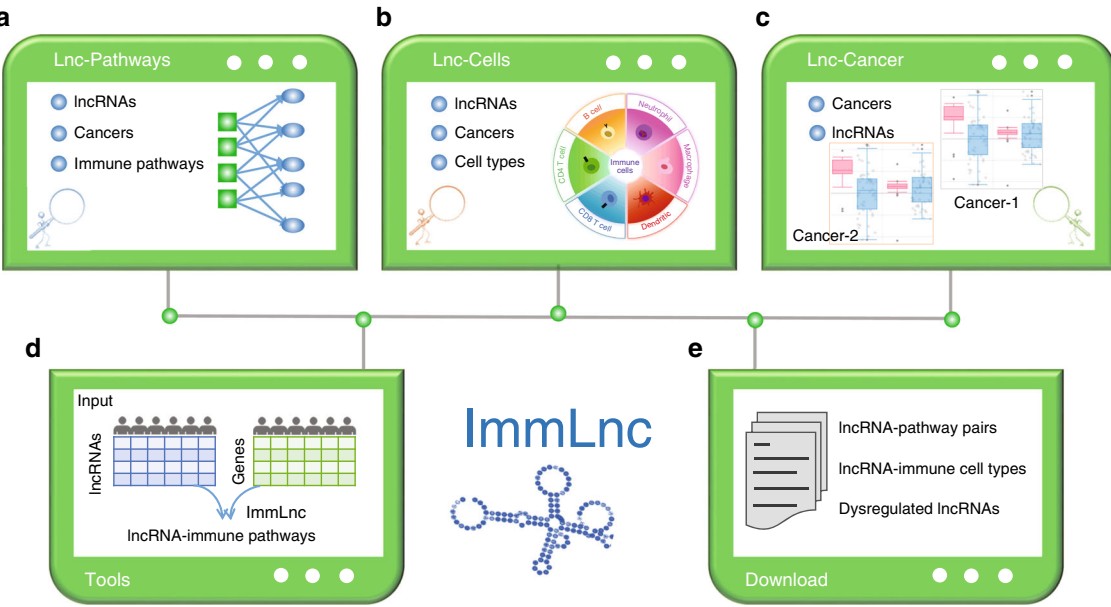

**Fig. 7 Diagram of the web-based ImmLnc resource. a** The query of immune pathway-related lncRNAs in cancer. **b** The query of immune cell infiltration-related lncRNAs across cancer types. **c** The query of cancer-related lncRNAs. **d** The ImmLnc package can be downloaded for local analysis based on paired lncRNA and mRNA expression. **e** All resources on this website can be downloaded for further analysis.

immune cell infiltration were likely to exhibit significantly higher expression in B cells and T cells (Supplementary Fig. 10). Moreover, we explored the tissue specificity of lncRNAs in different tumor tissues. First, we identified the tissue-enriched lncRNAs in each cancer type[49]. We found that the immune-related lncRNAs identified by the ImmLnc pipeline significantly overlap with tissue-enriched lncRNAs in 96.97% (32/33) of cancer types (Supplementary Fig. 23). These results suggest the immune-related lncRNAs exhibit high tissue-specific expression.

Although preliminarily applied to identify the regulators of immune-related pathways for further experimental validation, ImmLnc has already provided some biological insights. First, the predicted results indicate that not all immune-related pathways are equally associated with lncRNAs; cytokine and cytokine receptor pathways are likely to be correlated with more lncRNAs. These observations suggest that these two pathways might be more critical in cancer. Second, ImmLnc helps prioritizing the cancer-related lncRNAs and it identified several novel lncRNAs that may be potential targets in cancer. We also explored the expression ranges of immune-related lncRNAs and found that they were likely to be more highly expressed than other lncRNAs (Supplementary Fig. 24). In particular, the 28 lncRNAs identified in non-small cell lung cancer exhibit higher expression in >75% of patients. These results suggest that a proportion of lncRNAs can be detected at sufficiently robust expression levels to act as biomarkers. Third, based on the regulation of immune pathways and correlation with immune cell infiltration, we have demonstrated that ImmLnc can characterize different cancer subtypes (proliferative, intermediate, and immunological), which show distinct responses to chemotherapy. Finally, we focused on 17 immunologically relevant gene sets curated in Immport[17], which is one of the largest open repositories of immunology data. The ImmLnc pipeline can also be extended to other functional gene sets (such as essential genes) to identify the lncRNA regulators. Moreover, the idea of ImmLnc is not limited to dissecting lncRNA regulation and may be applied to identify other regulators (such as transcription factors and microRNAs) that are critical to immune-related pathways.

Identification of the targets of lncRNAs is a critical step to investigate their function. However, limited targets have been identified. In the present study, we identified the genes that are correlated with lncRNA expression and further investigated the enriched immune pathways. However, determining how a particular lncRNA influences gene expression remains a challenge. Many potential ways by which lncRNAs can exert their functions have been revealed, including acting as signal lncRNAs or scaffold lncRNAs[50]. With the development of RNA-centric methods, such as chromatin isolation by RNA purification[51] and capture hybridization analysis of RNA targets[52], we will be able to identify the interaction targets of lncRNAs in a native context. Identification of these targets will further improve our insight in the lncRNA-mediated immune pathway regulation and advance our understanding of lncRNA functions.

In summary, lncRNAs represent an additional layer of immune system complexity. It is important that we understand the function of lncRNAs in immune regulation. Continued investigation of the immune-related lncRNAs identified here will aid in the development of better immunotherapies for human cancer and other diseases.

## Methods

**Collection of immune-related genes**. Human immune-related genes were obtained from the ImmPort project[53]. These gene sets were widely used in immune-related studies[54,55]. All these genes are mapped to Ensembl IDs. In total, we obtained 1811 genes in 17 immune-related pathways for subsequent analyses.

**Genome-wide lncRNA and mRNA expression across cancer types**. Our analyses are mainly based on lncRNA and gene expression datasets generated by the TCGA Research Network (http://cancergenome.nih.gov/). In total, 33 different TCGA projects, each representing a specific cancer type, were analyzed. RNA-Seq-based gene expression profile data were obtained from the TCGA project via the R package "TCGAbiolinks"[56]. We downloaded the fragments per kilobase of transcript per million mapped reads-based gene expression and the raw read count for 33 types of cancer. Based on the gene annotations in GENCODE[57], we divided the gene expression profiles into lncRNA and protein-coding gene expression for each cancer type. The lncRNAs were further classified into different subtypes based on the classification in GENCODE. In total, 19,663 coding genes and 15,513 lncRNAs were included. Next, we excluded the coding genes and lncRNAs with zero reads in all samples. The expression values of lncRNAs and protein-coding genes were log-transformed. The clinical information of tumor patients, including the survival

status, stage, grade, survival time, and response to chemotherapy, was also downloaded from the TCGA project.

**ImmLnc: identification of immune-related lncRNAs in cancer**. To identify the potential lncRNA modulators of immune-related pathways, we proposed a computational method that integrates lncRNA and gene expression data (Fig. 1a). Briefly, all coding genes were ranked based on their correlation in expression with a specific lncRNA. The ranked gene list was subjected to each immune-related pathway to explore whether the immune genes are enriched in the top or bottom of the list. The lncRES score was calculated for each lncRNA–pathway pair. This process was repeated for all combinations of lncRNA and immune-related pathways. Based on a permutation test, all lncRNA–pathway pairs with significantly higher lncRES scores were identified in cancer.

Identification of lncRNA-correlated genes. For each lncRNA of interest, we first ranked all coding genes based on the correlation of their expression with this lncRNA. The expression of lncRNA $i$ and gene $j$ across tumor patients was defined as $L(i) = (l_1, l_2, l_3, ...l_i, ..., l_m)$ and $G(j) = (g_1, g_2, g_3, ..., g_j, ..., g_m)$, respectively. The tumor purity scores across $m$ patients are defined as $P = (p_1, p_2, p_3, ...,p_i, ..., p_m)$. We first calculated the partial correlation coefficient (PCC) between the expression of lncRNA $i$ and gene $j$ by considering the tumor purity as a co-variable, i.e.,

$$PCC(ij) = \frac{R_{LG} - R_{LP} * R_{GP}}{\sqrt{1 - R_{LP}^2} * \sqrt{1 - R_{GP}^2}}$$

where $R_{LG}$, $R_{LP}$, and $R_{GP}$ are the correlation coefficients between the expression of lncRNA $i$ and coding gene $j$, the expression of lncRNA $i$ and tumor purity, and the expression of gene $j$ and tumor purity, respectively. In addition, we obtained the $P$-value for the PCC, defined as $P(ij)$. For each lncRNA–gene pair, we calculated the rank score (RS) as follows:

$$RS(ij) = -log10(P(ij)) * sign(PCC(ij)).$$

All genes were ranked based on $RS$ scores and then subjected to enrichment analysis.

LncRNA modulators of immune-related pathways. Motivated by the idea of GSEA[15,16], we mapped the genes in each immune-related pathway to the ranked gene list. Next, we calculated the enrichment score (ES) based on the GSEA. If there were $N$ genes in the ranked gene list $L = \{g_1, g_2, g_3, ..., g_N\}$, the ranked score was $RS(g_j) = r_j$. We first evaluated the fraction of genes in pathway $H$ ("hits") weighted by their RS and the fraction of genes not in $S$ ("misses") present up to a given position $i$ in $L$, as follows:

$$P_{hit}(H, i) = \sum_{\substack{g_i \in H \\ j \leq i}} \frac{|r_j|^p}{N_R}, \text{ where } N_R = \sum_{g_i \in H} |r_j|^p,$$

$$P_{miss}(H, i) = \sum_{\substack{g_j \notin H \\ j \leq i}} \frac{1}{(N - N_I)}.$$

The ES score was the maximum deviation from zero of $P_{hit} - P_{miss}$. In addition, a $P$-value was calculated for each pathway that includes $N_I$ genes as follows:

$$p(ES(N, N_I) < ES_{ik}) = \sum_{q=-\infty}^{\infty} (-1)^q \exp(-2q^2 ES_{ik}^2 n),$$

$$n = \frac{(N - N_I)N_I}{N},$$

where $ES_{ik}$ is the ES score between lncRNA $i$ and immune pathway $k$, $N$ is the number of genes in the ranked list, and $N_I$ is the number of genes in the specific immune pathway. $P$-values were adjusted using the FDR. Moreover, following a previous study[58], we combined the $P$-value and the $ES$ score to an *lncRES* score, i.e.,

$$lncRES(i, k) = \begin{cases} 1 - 2p; & \text{if } ES(ik) > 0, \\ 2p - 1; & \text{if } E(ik) < 0. \end{cases}$$

Thus, the lncRES scores ranged from −1 to 1. We considered the lncRNA–pathway pairs with the absolute lncRES scores >0.995 and FDR <0.05 as significant ones.

**Prioritization of cancer-related lncRNAs**. To identify cancer-related lncRNAs, we first obtained the literature-curated cancer lincRNAs from Lnc2Cancer v2.0[59] and LncRNADisease 2.0[60]. Here we focused on the lincRNAs. Our hypothesis is that the cancer-related lncRNA-mediated immune pathway regulation is likely to be present in more cancer types and with other lncRNAs. Thus, we first ranked each lincRNA based on the number of cancer types in which it regulates immune-related pathways. Each pathway was analyzed separately. For each lincRNA, we obtained 17 relative ranks based on the immune pathways. Next, we also ranked the lincRNAs based on the number of cancer types in which they showed expression perturbations. Finally, the lincRNAs were ranked based on the average of 18 relative RSs, which was defined as

$$F(\text{rank}) = \frac{\sum_{i=1}^{17} R_i + R_{\text{deg}}}{18},$$

where $R_i$ is the normalized rank based on the immune pathway and $R_{\text{deg}}$ is the normalized rank based on the differential expression analysis. The relative RSs between cancer-related lincRNAs and others were compared based on the one-sided Wilcoxon's rank-sum test.

**Classification of patients based on immune-related lncRNAs**. To identify the lncRNAs that might be used for cancer subtyping, we first identified the lncRNAs that are correlated with infiltration of all six immune cells in two types of lung cancer (LUAD and LUSC). These lncRNAs were mapped to immune lncRNA regulators in each cancer type and the overlapping lncRNAs in both cancer types were used to classify the lung cancer patients.

We next used the ConsensusClusterPlus R package to identify the optimum number of clusters in pan-lung cancer lncRNA expression data[61]. The lncRNA expression profiles were first normalized by the Z-score and then subjected into analysis using this R package. We used 100 iterations and 80% of the samples were resampled, and $k$ ranged from 2 to 10 using hierarchical clustering with Pearson's correlation as the similarity metric. We selected a preferred clustering result by considering the relative change in area under the Cumulative Distribution Function curve[62]. Moreover, we obtained the public multiplatform-based molecular subtypes of lung cancer from recent literature[63], including the classifiers based on DNA copy number variation, DNA methylation, miRNA and gene expression, protein expression, and Cluster-Of-Cluster-Assignments-based results. The differentiation and pathway activity scores for each patient were obtained from one recent study[63]. The stemness scores for cancer patients were downloaded from the study of Malta et al.[64] and we used the DNA methylation and gene expression-based scores.

**Reporting summary**. Further information on research design is available in the Nature Research Reporting Summary linked to this article.

## Data availability

All accession codes, unique identifiers, or web links for publicly available datasets are described in the paper. All data supporting the findings of the current study are listed in Supplementary Data 1–6, Supplementary Figs. 1–24, and our online data portal (http://bio-bigdata.hrbmu.edu.cn/ImmLnc). The source data underlying Figs. 1b–d, 2a–c, 3a–g, 4d–f, 5b–f, and 6a–b, e are provided as a Source Data file.

## Code availablity

All codes are available upon reasonable request.

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

## Acknowledgements

This work was supported by the National Key R&D Program of China (2018YFC2000100); the National Natural Science Foundation of China (61873075, 31871338, and 31970646); Heilongjiang Touyan Innovation Team Program; and Natural Science Foundation for Distinguished Young Scholars of Heilongjiang Province (JQ2019C004).

## Author contributions

X.L., J.X., and Y.L. conceived of the project. T.J., W.Z., and Y.L. designed and performed the research with contributions from X.L., Q.W., J.X., J.Y., and L.C. J.L. and W.Z. provided constructive feedback and constructed the web-based resource. Y.Z. and Y.L. analyzed the single-cell data. J.X., Y.L., and X.L. supervised research and provided critical advice on the study. Y.L., J.X., and T.J. wrote the manuscript, with input from other co-authors.

## Competing interests

The authors declare no competing interests.
