## [Peer Review File · Nature Communications]

Reviewers' comments:

Reviewer #1 (Remarks to the Author):

In this study, the authors developed an integrated algorithm, ImmLnc, for identifying lncRNA regulators of immune-related pathways. By applying the ImmLnc algorithm to TCGA data, the authors identified immune relevant lncRNA in various cancers. Moreover, the study reveals the connection between predicted lncRNAs expression and immune infiltration in multiple cancer subtypes. The predicted lncRNA signature was able to categorize the pan-lung cancer into three molecular subtypes with different immune infiltration and survival status.

Overall, the ImmLnc pipeline will be a valuable resource for understanding immune relevant lncRNA function and for advancing identification of immunotherapy targets. Thus far, only a few immune-related lncRNAs associated with the immune system have been identified and even fewer validated. Therefore, it is important to first identify immune-related lncRNAs that can be exploited as biomarkers or targets for immunotherapy. If accurate, the ImmLnc could open the doors for researchers and industry by providing "valuable resource in the elucidation of precision medicine." Overall, this is a well-executed study, but a few queries remain

Comments:

1. This reviewer has strong reservations about the methods of identifying these lncRNAs and labeling them as "regulators of immune-related pathways" and relevance for "identification of immunotherapy targets" in this study. Overall, the authors provide evidence that they identified "cancer-related" lncRNAs that may have some association with cytokines but do not achieve their stated purpose. The validation data provided are inconsistent with what is already known about immune infiltration and make-up of tumors from the same datasets, which raises doubt that the lncRNAs identified are truly correlated with the immune system. Theoretically, ImmLnc could be a highly utilized and important resource, so it is critical that the pipeline is accurate.
2. All past studies identifying lncRNAs deemed to be "regulators of immune-related pathways," including those that are candidates as "immunotherapy targets" and many cited in this study, were identified and characterized in immune cells. The identification of the well-validated immune signatures, used in this study and others, was done in immune cells but then applied to bulk tumor data, not the other way around. However, the authors seem to focus on identifying lncRNAs expressed in tumor cells that may be correlated with broad immune cell signatures, which do not achieve the specific objectives of the author's stated purpose and is significantly less novel because tumor-expressed lncRNAs have been studied in large scale in multiple published studies.
3. It is unclear why the authors did not derive or validate "immune lncRNAs" to widely available public sequencing data on various immune cell subsets and then apply to tumor data. They should, at the minimum, confirm that a majority of lncRNAs identified through their pipeline are expressed in immune cell populations. This is especially true as these correlations were derived from immune cell signatures ("cytokines" or "cytokine receptors") derived from data in immune cells.
4. What is the rationale of focusing specifically on the 17 immunologically-relevant gene sets in this study?
5. What is the total number of lncRNAs identified in each cancer type? Is there any correlation between total number of lncRNAs and number of predicted immune associated lncRNAs in different cancers?
6. It is well known that lncRNAs are expressed in a highly cell/lineage-specific manner. Several studies have already reported that lncRNAs are specifically expressed in immune cell types, and the authors have identified immunologically relevant lncRNAs in the study. The authors should

discuss the expression specificity of their immune associated lncRNAs in immune and tumor cells.

7. Based on immune-related lncRNA groups, authors predicted 3 subtypes in lung cancer (C1, C2, C3). While the authors showed the differences between C1 and C2 groups, the differences or similarities between the C2 and C3 groups were not discussed in-depth.

8. Fig. 1 - "Cytokines (7)" and "Cytokine Receptors (8)" should be "umbrella," generalized categories which include almost all of the other categories (at least, 5-6, 9-17). To say that lncRNAs correlate with "cytokines" but not any of the other categories, such as "interferons" or "interleukins" (Supplementary figure 1b), is suspect. As "cytokines" may or may not be immune cell-related (i.e. cytokines expressed by adipose tissue, fibroblasts, that have nothing to do with anti-tumor immunity, etc.) it is unclear if these associated lncRNAs would have any relevance to immunotherapy.

9. Fig 2 - The authors have named individual lncRNAs, such as MIAT and PVT1, that are upregulated as well as are known to be associated with tumor survival/proliferation and associated with "cytokines." This is unhelpful, as this merely reiterates that these are tumor lncRNAs and may be related to any number of cytokines. This could have been strengthened by the additional validation. However, the authors do not validate the same lncRNAs by correlating them with specific cytokines pathways (i.e. downregulation of IFN γ , cytotoxic T cell transcriptomic signature, etc.), which should be very simple to do with the data. Instead, they select a different set of lncRNAs to correlate with CD8 T cell infiltration. For these lncRNAs, please provide transcriptomic validation data similar to the rest of the paper. If these are not correlated with T cell infiltration, then what other specific and cancer-relevant immune cell signatures are they correlated with to support the authors' claim that their associations with "cytokines" are relevant ones?

10. Fig 2 - It is reassuring that the group of significant lncRNAs for a given cancer type have high correlation across different data sets. However, I disagree with the claim that the algorithm can recapitulate "immune-related" lncRNAs; it is able to recapitulate the same set of lncRNAs, and therefore it is reliable but still may be inaccurate in identifying true immune-correlated genes relevant to precision medicine.

11. Fig 2 - Again, the authors do a decent job of demonstrating whether or not lncRNAs play an important role in cancer --which is known—but then overstep by asserting evidence of that these are "perturbed immunology regulators" where there is none. It is suspect that cancer types with similar tissue origins that share the same immune-related lncRNAs and "tissue-of-origin" are not predictive of the type of immune cell infiltration or immune response. This supports the notion that the pipeline is actually identifying tumor-specific lncRNAs, which may or may not be associated with cytokines with questionable relevance to the immune response.

12. Fig 3a-g - If there is supposed to be a correlation between lncRNA expression and the number of each immune cell subtype, it is hard to believe that such a high proportion of immune cells in the tumor would be dendritic cells, neutrophils, or B cells for any cancer. Additionally, prostate cancer (PRAD), which is known to be poorly immunogenic and have low T cell infiltration, seems to have one of the highest estimates of CD8+ T cell infiltration based on the lncRNA data.

13. Fig. 4C: Based on the ImmLnc method, authors identified the top-ranked 10 lncRNAs. Among these 10 lncRNAs, LINC00944 and RP11-325F22.2 can regulate the TCR signaling pathway activity in different cancer types. Are all other genes in the top-ranked list that are also associated with TCR signaling?

14. Fig 5-6 - Classification of molecular or immunologic subtypes of cancer by immune-associated lncRNAs is very interesting and translationally relevant. However, these are only relevant as next steps in a reliable pipeline after the lncRNAs identified by ImmLnc have been validated. This validation could include 1) correlation with functionally relevant cancer-related immune signaling

pathways, 2) showing that the lncRNAs are largely specifically expressed in immune cells, and that 3) they were additionally expressed by both tumor cells/immune cells in RNA-seq bulk tumor data as stated previously.

15. Fig. 5E - The authors claim that patients in the C1 group have significantly poorer prognosis than other patients. However, the difference is too small to claim the significance.

16. Fig. 6B - Authors compared the HRD scores across different subtypes and found that patients in the C2 subtype were with significantly higher HDR scores. However, this bar chart does not support their claim. There were very low/no differences in HDR scores between C1 and C2 groups.

17. In Fig 1, the title of section 2, "lncRNA Regulators are Likely to Expression Perturbation in Cancer" is grammatically incorrect.

Reviewer #2 (Remarks to the Author):

Overall summary

In this manuscript, Li et al develop a pipeline "ImmLnc" to identify lncRNAs that may regulate the immune response in cancer, by identifying lncRNAs that correlate with specific immune gene expression signatures and immune cell infiltration. The authors used this approach to examine non-small cell lung cancer more closely, and were able to identify three subgroups of lung cancer with different intrinsic features (e.g. mutational burden) and differing immune responses.

Overall the approach taken in this manuscript appears to be largely sound and may be a useful methodology to apply when investigating lncRNAs in cancer, with clearly presented figures and most data/methods described sufficiently. However, the manuscript contains a large number of grammar errors that make it difficult to understand results/methodology at certain points, and would benefit from a substantial rewriting of the text. Furthermore, there are several experimental/analytical points that should be addressed before publication.

Major Points

1. The manuscript would greatly benefit from further proofreading and/or rewriting to improve the English throughout.
2. One key question not addressed is whether the lncRNAs identified as correlating with immune responses are derived from host cells or infiltrating immune cells. Indeed, there is no data presented anywhere in the manuscript derived from pure immune cell populations. Given that there are published RNA-seq datasets available for the majority of the immune cells identified as tumour-infiltrating cells, these could be examined to identify for example which lncRNAs correlating with CD8 T cell infiltration are specifically expressed in CD8 T cells and not the host tumour.
3. The authors take a very broad correlative approach in this study, however some consideration should be given to the fact that most lncRNAs are expressed at very low levels. This is particularly pertinent when considering (following on from point 2) that many of the identified lncRNAs may be derived from tumour-infiltrating cells that may make up a relatively small fraction of the overall tumour. Can the authors provide some further evidence that a proportion of the lncRNAs at least can be detected at sufficiently robust expression levels to act as biomarkers?
4. lncRNAs are broadly grouped together throughout the manuscript with no subclass analysis made, apart from after line 193 when an unexplained switch to intergenic lncRNAs is made. One might expect that certain subclasses of lncRNAs (e.g. enhancer RNAs, promoter-associated lncRNAs) to naturally correlate more with the expression of proximal genes in the immune response. The authors should try and address whether there are subclasses of lncRNAs that are more likely to be correlative than others.
5. Some of the conclusions in the last section of the manuscript (figures 5 and 6) seem somewhat over-interpreted in places. Some more specific points are raised below, but whilst the C1 cluster

appears immunologically distinct from the rest of the cases, I am not convinced that there is any real differences between the C2 and C3 classes.

6. Although the methods generally contain sufficient detail, some more explanation of key analyses should be made briefly in certain places in the manuscript and/or figure legends to aid the readers with understanding the points being made.

Minor points

The following points are broken down by lines in the manuscript:

44 – This is a somewhat contentious statement that derives from early studies into the lncRNA field, now widely debunked (see Graur et al, 2013, Genome Biol Evol for example) . It may be truer if aggregating all possible cell types, but in a given cell type at any one time the majority of the genome will remain untranscribed. I would remove or soften this statement.

86- define/describe RS score in manuscript. Some further description of the correlation scores and GSEA analysis would be helpful too as it's a key point in the analysis.

125-128 – it would be good to know the no. of lncRNA-pathways relationships that did not overlap between matched datasets (i.e. provide percentage figures).

139 – In supplementary figure 2d, DLBCL and AML are highlighted as similar – I'd question whether these are really similar at all – these are very different cancers.

148 – It would be good to see the identification of immune infiltrating subsets validated by an orthogonal approach e.g. CIBERSORT.

162-164 – I do not understand the point being made in this sentence.

193 – Where do these 5050 lincRNAs derive from? Why the switch to intergenic lncRNAs only?

238-239 – how were the tumour differentiation, hippo pathway, stem and cell cycle activity scores calculated? I cannot see these described in the methodology

242 -244 – the differences in survival between groups appears relatively modest, however this may be exacerbated by the extreme length of the follow up and its impact on the curve. It might be worth considering how many patients are actually alive after say 3000 or 4000 days, and whether any events after this point are relevant to the cancer diagnoses (i.e. by the point patients may be succumbing to old age).

259 – HRD scores for C2 higher than C3 but not C1 as implied

261-263 – where are these 160 signatures derived from?

265-269. What about C3 group? These are also generally high in B cell signatures at least and should be discussed.

277-278 – Differences in response are very modest, particularly in comparison to C3 group. Can the authors provide a kaplan meier curve analysis instead?

323-326 – having new data appear in the discussion is confusing and should be mentioned earlier in the results section or be removed.

416-421- The description of identifying differentially expressed lncRNAs is confusing and does not sound stringent e.g. defining a lncRNA as being expressed if it is >0 is a very low threshold

726-728 – Figure 3 legend – it is not clear to me what is being shown in the inset bar-graph –

please reword description

744 – 746 – Figure 5 legend – please provide more detail on the calculated scores.

Reply to Reviewer #1:

In this study, the authors developed an integrated algorithm, ImmLnc, for identifying lncRNA regulators of immune-related pathways. By applying the ImmLnc algorithm to TCGA data, the authors identified immune relevant lncRNA in various cancers. Moreover, the study reveals the connection between predicted lncRNAs expression and immune infiltration in multiple cancer subtypes. The predicted lncRNA signature was able to categorize the pan-lung cancer into three molecular subtypes with different immune infiltration and survival status.

Overall, the ImmLnc pipeline will be a valuable resource for understanding immune relevant lncRNA function and for advancing identification of immunotherapy targets. Thus far, only a few immune-related lncRNAs associated with the immune system have been identified and even fewer validated. Therefore, it is important to first identify immune-related lncRNAs that can be exploited as biomarkers or targets for immunotherapy. If accurate, the ImmLnc could open the doors for researchers and industry by providing “valuable resource in the elucidation of precision medicine.” Overall, this is a well-executed study, but a few queries remain.

Reply: Many thanks for appreciating our study. The manuscript has been greatly improved by addressing the reviewers' comments.

Comments:

1. This reviewer has strong reservations about the methods of identifying these lncRNAs and labeling them as “regulators of immune-related pathways” and relevance for “identification of immunotherapy targets” in this study. Overall, the authors provide evidence that they identified “cancer-related” lncRNAs that may have some association with cytokines but do not achieve their stated purpose. The validation data provided are inconsistent with what is already known about immune infiltration and make-up of tumors from the same datasets, which raises doubt that the lncRNAs identified are truly correlated with the immune system. Theoretically, ImmLnc could be a highly utilized and important resource, so it is critical that the pipeline is accurate.

Reply: Thanks for pointing this out. Because a very limited number of immune-related lncRNAs have been identified, it is difficult to validate the accuracy of this pipeline directly. Therefore, we validated the accuracy of the ImmLnc pipeline in the following way. **(1)** We found that the expression of the identified lncRNAs is likely to be perturbed across cancer types (Fig. 2d). **(2)** The expression of these lncRNAs is correlated with immune cell infiltration in cancer (Fig. 3). **(3)** The ImmLnc pipeline helps prioritizing cancer-related lncRNAs and cancer subtyping (Figs. 4 and 5).

Moreover, we have accepted the suggestion and provided more evidence to validate the ImmLnc pipeline in our revised manuscript. **(4)** We found that the immune-related lncRNAs are likely to exhibit significantly higher expression in immune cell populations (Supplementary Figs. 6–8). **(5)** Immune-related

lncRNAs are more likely to co-occur with “immune” in the literature (Supplementary Fig. 22). **(6)** The immune-related lncRNAs are expressed in a highly tissue-specific manner (Supplementary Fig. 23). **(7)** We found that ImmLnc can capture experimentally validated lncRNAs that potentially regulate the functional pathway based on CRISPR-Cas9 data (Supplementary Fig. 21). **(8)** The lncRNA–pathway associations across cancer types significantly overlap with those identified in immune cell populations (Supplementary Table 3).

Collectively, all these results suggest that the lncRNAs identified by ImmLnc play crucial roles in cancer immunology. The detailed results are shown in the revised manuscript as well as in the following point-by-point response.

2. All past studies identifying lncRNAs deemed to be “regulators of immune-related pathways,” including those that are candidates as “immunotherapy targets” and many cited in this study, were identified and characterized in immune cells. The identification of the well-validated immune signatures, used in this study and others, was done in immune cells but then applied to bulk tumor data, not the other way around. However, the authors seem to focus on identifying lncRNAs expressed in tumor cells that may be correlated with broad immune cell signatures, which do not achieve the specific objectives of the author’s stated purpose and is significantly less novel because tumor-expressed lncRNAs have been studied in large scale in multiple published studies.

Reply: Many thanks for this comment. In this study, we integrated paired lncRNA and mRNA expression profiles to identify immune-related lncRNAs in cancer. Ideally, it is better to identify lncRNAs based on expression profiles across immune cells. However, the number of samples for currently available immune datasets is limited, which might limit the accuracy of the proposed pipeline. Moreover, the majority of currently available immune datasets were obtained from peripheral blood. It is not known whether the immune regulation in blood is the same as in tumor tissues. Therefore, we applied the pipeline to pan-cancer data from TCGA. In our pipeline, tumor purity was considered as a co-variable. Based on the indirect evidence presented in our manuscript (points 1–8 listed in comment 1), we think ImmLnc can identify lncRNAs that potentially regulate the immune-related pathways. Moreover, we have accepted the suggestion and applied the ImmLnc pipeline to two datasets across immune cell populations (details in Supplementary methods). We found that the lncRNA–pathway pairs significantly overlap with those identified in cancer tissue datasets (Supplementary Table 3). These results suggest that ImmLnc can identify potential regulators of immune-related pathways. We have discussed this issue in our revised manuscript.

3. It is unclear why the authors did not derive or validate “immune lncRNAs” to widely available public sequencing data on various immune cell subsets and

then apply to tumor data. They should, at the minimum, confirm that a majority of lncRNAs identified through their pipeline are expressed in immune cell populations. This is especially true as these correlations were derived from immune cell signatures (“cytokines” or “cytokine receptors”) derived from data in immune cells.

Reply: Thanks for this comment. We have accepted the suggestion and analyzed ten single-cell sequencing data downloaded from PanglaoDB (Oscar *et al.*, 2019, Database). We found that a significantly higher proportion of immune-related lncRNAs is expressed in immune cells across cancer types (Supplementary Fig. 6). Second, we derived the expression of lncRNAs in immune cells of TCGA samples based on the ideas from RESPECTEx (Ng *et al.*, 2019, *Nucleic Acids Research*). We found that these lncRNAs were significantly more highly expressed in immune cells (Supplementary Fig. 7). We also investigated the expression of lncRNAs in immune cell populations, which were identified by single-cell sequencing in lung cancer (Diether *et al.*, 2018, *Nature Medicine*). We found that the identified immune-related lncRNAs in lung cancer exhibited significantly higher expression than other lncRNAs in B cell and T cell populations (Supplementary Fig. 8). Together, these results further suggest that the identified lncRNAs are likely to be highly expressed in immune cells. We have provided these results in our revised manuscript.

Supplementary Fig. 6. Heat map showing odd ratios of Fisher's exact test. The proportions of immune lncRNAs and nonimmune lncRNAs expressed in immune cell populations were compared in 10 single-cell sequencing datasets. All $P < 0.001$.

Supplementary Fig. 7. The enrichment of immune-related lncRNAs expressed in immune cell populations. The y-axis shows the $\log_2(\text{ratio})$ between average expression of immune-related lncRNAs and other lncRNAs in immune cells. Green indicates the P -values for Wilcoxon rank sum test were less than 0.01.

Supplementary Fig. 8. The expression of immune-related lncRNAs in immune cell populations. The top panels show lncRNAs identified in LUAD, and the bottom panels show lncRNAs identified in LUSC. $***P < 0.001$, Wilcoxon rank sum test.

4. What is the rationale of focusing specifically on the 17 immunologically-relevant gene sets in this study?

Reply: Many thanks for this question. Recent advances have identified a

number of genes associated with immunology. In our study, we focused on the gene sets curated in ImmPort, which is one of the largest open repositories of immunology data (Bhattacharya et al., 2018, *Sci Data*). This dataset have been widely used in a number of immune-related studies (Li et al., 2017, *JAMA Oncology*; Cui et al., 2018, *Clinical Cancer Research*; Shen et al., 2019, *EbioMedicine*). Taking advantage of the enriched ImmPort datasets deposited by large consortia and individual labs in the immunology community, we identified the lncRNAs that were potentially associated with these gene sets. In addition, our method can be easily extended to other functional gene sets. We have discussed this in the Discussion section of our revised manuscript.

5. What is the total number of lncRNAs identified in each cancer type? Is there any correlation between total number of lncRNAs and number of predicted immune associated lncRNAs in different cancers?

Reply: Thank you very much for pointing this out. Our inappropriate and unclear description might confuse the readers. In the lncRNA expression profile obtained from RNA-Seq, the total number of lncRNAs among all cancer types was the same. We first filtered out lncRNAs with zero expression in > 30% of the samples. We provided the proportion of immune-associated lncRNAs in Fig. 1d in our original manuscript. To make this clearer, we calculated the proportion as the number of immune-associated lncRNAs divided by the number of expressed lncRNAs in each cancer type. We have updated Fig. 1d in our revised manuscript. In addition, we have accepted the suggestion and calculated the correlation coefficient between the number of expressed lncRNAs and immune-associated lncRNAs. We found that they were strongly correlated (Supplementary Fig. 1). We have discussed this in our revised manuscript.

Fig. 1d. The number of immune-related lncRNAs identified in each cancer type. The top y-axis shows the number of lncRNAs and the bottom y-axis shows the proportion of lncRNAs.

Supplementary Fig. 1. The number of immune-related lncRNAs and expressed lncRNAs in different cancer types. The x-axis shows the number of expressed lncRNAs in each cancer type, and the y-axis shows the number of immune-related lncRNAs identified by the ImmLnc pipeline. Each dot represents one cancer type. The Pearson Correlation Coefficient (PCC) is 0.60 and $P = 0.0002$.

6. It is well known that lncRNAs are expressed in a highly cell/lineage-specific manner. Several studies have already reported that lncRNAs are specifically expressed in immune cell types, and the authors have identified immunologically relevant lncRNAs in the study. The authors should discuss the expression specificity of their immune associated lncRNAs in immune and tumor cells.

Reply: Many thanks for this suggestion. We have accepted the suggestion and explored the tissue specificity of lncRNA expression. First, we identified the tissue-enriched lncRNAs in each cancer type based on previous methods (Mathias et al., Science, 2015; Mathias et al., Molecular Systems Biology, 2016). We found that the immune-related lncRNAs significantly overlap with tissue-enriched lncRNAs in 96.97% (32/33) of the cancer types (Supplementary Fig. 23). These results suggest that the immune-related lncRNAs are expressed in a highly tissue-specific manner.

Supplementary Fig. 23. Tissue specificity of immune-related lncRNAs. Each dot represents one cancer type; the x-axis represents the O/E value, and the y-axis represents the $-\log_{10}(P\text{-value})$.

7. Based on immune-related lncRNA groups, authors predicted 3 subtypes in lung cancer (C1, C2, C3). While the authors showed the differences between C1 and C2 groups, the differences or similarities between the C2 and C3 groups were not discussed in-depth.

Reply: Thanks for this suggestion. We have discussed C2 and C3 patients in-depth in our revised manuscript. We found that there are significant differences in tumor differentiation, cell cycle activity, mutation burden, HRD score, and T cell infiltration between C2 and C3. All these results are discussed in our revised manuscript.

8. Fig. 1 - “Cytokines (7)” and “Cytokine Receptors (8)” should be “umbrella,” generalized categories which include almost all of the other categories (at least, 5-6, 9-17). To say that lncRNAs correlate with “cytokines” but not any of the other categories, such as “interferons” or “interleukins” (Supplementary figure 1b), is suspect. As “cytokines” may or may not be immune cell-related (i.e. cytokines expressed by adipose tissue, fibroblasts, that have nothing to do with anti-tumor immunity, etc.) it is unclear if these associated lncRNAs would have any relevance to immunotherapy.

Reply: Many thanks for this comment. Our inappropriate and unclear description might confuse the readers. In this supplementary figure, we show the number of lncRNAs that were likely to regulate corresponding immune-related pathways across 33 cancer types. We found that a higher number of lncRNAs for “Cytokines/Cytokines receptors” pathways. We have accepted the suggestion and calculated the overlap of lncRNAs that were

likely to regulate “Cytokines/Cytokines receptors” and “interferons/interleukins”. Although several lncRNAs can regulate both “Cytokines/Cytokines receptors” and “interferons/interleukins” pathways, a number of lncRNAs only regulate one of the two. We further investigated the lncRNAs that only regulate “Cytokines/Cytokines receptors” pathways. We also found that a significantly high proportion of lncRNAs co-occur with “immune” in the literature (Supplementary Fig. A1). This suggests that these associated lncRNAs are likely to be involved in immunology.

Supplementary Fig. A1. The odds ratio distribution across cancer types. Fisher’s exact test was used to test whether the immune-related lncRNAs were more likely to co-occur with “immune” in the literature than other lncRNAs. The pie chart shows the proportion of cancer types with OR > 1 and P < 0.001.

9. Fig 2 - The authors have named individual lncRNAs, such as MIAT and PVT1, that are upregulated as well as are known to associated with tumor survival/proliferation and associated with “cytokines.” This is unhelpful, as this merely reiterates that these are tumor lncRNAs and may be related to any number of cytokines. This could have been strengthened by the additional validation. However, the authors do not validate the same lncRNAs by correlating them with specific cytokines pathways (i.e. downregulation of IFN γ , cytotoxic T cell transcriptomic signature, etc.), which should be very simple to do with the data. Instead, they select a different set of lncRNAs to correlate with CD8 T cell infiltration. For these lncRNAs, please provide transcriptomic validation data similar to the rest of the paper. If these are not correlated with T cell infiltration, then what other specific and cancer-relevant immune cell signatures are they correlated with to support the authors’ claim that their associations with “cytokines” are relevant ones?

Reply: Many thanks for this comment. We have accepted the suggestion and

obtained the specific cytokines pathway scores (“IFNG_score” and “Cytotoxic cells”) from one recent publication (Thorsson, *et al.*, *Immunity*, 2018). We found that expression of the lncRNAs *MIAT* and *PVT1* was also correlated with the activities of these pathways in the majority of cancer types (Supplementary Fig. 4). In addition, we calculated the correlation between the expression of these lncRNAs and immune cell infiltration across cancer types. We found that the expression of *PVT1* and *MIAT* was significantly correlated with CD8 T cell infiltration in several cancer types (Supplementary Fig. 11), such as UVM and THCA for *PVT1* and SKCM and CHOL for *MIAT*. Together, these results suggest that these lncRNAs play critical roles in immunology.

Supplementary Fig. 4. The correlation between lncRNA expression and cytokine-related pathway activities in cancer.

Supplementary Fig. 11. The correlation between expression of lncRNAs and immune cell infiltration. The x-axis represents the Spearman Correlation Coefficient (SCC), and the y-axis represents the $-\log_{10}(P\text{-value})$. Each dot represents an immune cell type in one cancer. a, *PVT1*; b, *MIAT*.

10. Fig 2 - It is reassuring that the group of significant lncRNAs for a given cancer type have high correlation across different data sets. However, I disagree with the claim that the algorithm can recapitulate “immune-related” lncRNAs; it is able to recapitulate the same set of lncRNAs, and therefore it is reliable but still may be inaccurate in identifying true immune-correlated genes relevant to precision medicine.

Reply: Thanks for this comment. As there are currently no golden standard immune-related lncRNAs, we explored to what extent these lncRNAs co-occurred with “immune” in PubMed. We found that a significantly high proportion of immune-related lncRNAs co-occurred with “immune” (Supplementary Fig. 22). In addition, recent studies suggested that genes whose expression is negatively correlated with tumor purity and positively correlated with immune cell infiltration are likely to play important roles in immunology (Li et al., 2016, *Genome Biology*; Ng et al., 2019, *Nucleic Acids Research*). Therefore, we first identified these lncRNAs and calculated the overlap with the lncRNAs identified by ImmLnc. We found that a significantly higher proportion of immune-related lncRNAs across cancer types (Supplementary Fig. 12). These results provide additional evidence that the lncRNAs identified by the ImmLnc pipeline play critical roles in immunology. We have discussed this issue in our revised manuscript.

Supplementary Fig. 22. The odds ratio distribution in cancer types for comparison of co-occurrence with “immune” in the literature. Fisher’s exact test was used to test whether the immune-related lncRNAs were more likely to co-occur with “immune” in the literature than other lncRNAs.

Supplementary Fig. 12. The overlap of immune-related lncRNAs with lncRNAs negatively correlated with tumor purity and positively correlated with immune cell infiltration.

11. Fig 2 - Again, the authors do a decent job of demonstrating whether or not lncRNAs play an important role in cancer --which is known—but then overstep by asserting evidence of that these are “perturbed immunology regulators” where there is none. It is suspect that cancer types with similar tissue origins that share the same immune-related lncRNAs and “tissue-of-origin” are not predictive of the type of immune cell infiltration or immune response. This supports the notion that the pipeline is actually identifying tumor-specific lncRNAs, which may or may not be associated with cytokines with questionable relevance to the immune response.

Reply: Many thanks for pointing this out. In our revised manuscript, we have accepted the suggestion and added more evidence to validate the association between lncRNAs and immunology. For example, we found that the identified lncRNAs exhibited higher expression in immune cells (response to comment 3), were likely to co-occur with “immune” in the literature (response to comment 10), and significantly overlap with results based on immune cell data (response to comment 2). These results suggest that these lncRNAs are potentially associated with immunology. Moreover, we checked whether the lncRNAs in Fig. 2 were associated with cytokines. We found that they were likely to co-occur with “cytokine” in the literature (Supplementary Fig. 5). These results suggest that they were potentially associated with cytokines. We have discussed this issue in our revised manuscript.

Supplementary Fig. 5. The proportion of lncRNAs that co-occurred with “cytokine” in the literature. Fisher’s exact test was used to evaluate the difference.

12. Fig 3a-g - If there is supposed to be a correlation between lncRNA expression and the number of each immune cell subtype, it is hard to believe that such a high proportion of immune cells in the tumor would be dendritic cells, neutrophils, or B cells for any cancer. Additionally, prostate cancer (PRAD), which is known to be poorly immunogenic and have low T cell infiltration, seems to have one of the highest estimates of CD8+ T cell infiltration based on the lncRNA data.

Reply: Thank you very much for this comment. Our inappropriate and unclear description might confuse the readers. In this figure, the proportion was not calculated for immune cells but for lncRNAs. In Fig. 3a, the proportions were calculated as the number of immune-related lncRNAs that were correlated with immune cell infiltration level divided by the total number of immune cell infiltration-related lncRNAs. We have provided the details in the Methods section of our revised manuscript.

13. Fig. 4C: Based on the ImmLnc method, authors identified the top-ranked 10 lncRNAs. Among these 10 lncRNAs, LINC00944 and RP11-325F22.2 can regulate the TCR signaling pathway activity in different cancer types. Are all other genes in the top-ranked list that are also associated with TCR signaling?

Reply: Many thanks for this comment. These lncRNAs were ranked based on the number of cancer types in which the lncRNA–pathway correlation was observed. These lncRNAs were associated with TCR signaling in 3–27 cancer types. To make this result clearer, we have provided a supplementary figure (Supplementary Fig. 17) showing the correlation among lncRNAs and

pathways in our revised manuscript.

Supplementary Fig. 17. River plot showing the association between lncRNAs and immune-related pathways. The weight of the edges corresponds to the number of cancer types showing this association.

14. Fig 5-6 - Classification of molecular or immunologic subtypes of cancer by immune-associated lncRNAs is very interesting and translationally relevant. However, these are only relevant as next steps in a reliable pipeline after the lncRNAs identified by ImmLnc have been validated. This validation could include 1) correlation with functionally relevant cancer-related immune signaling pathways, 2) showing that the lncRNAs are largely specifically expressed in immune cells, and that 3) they were additionally expressed by both tumor cells/immune cells in RNA-seq bulk tumor data as stated previously.

Reply: Many thanks for this suggestion. As the reviewer suggested, the lncRNAs we used for classification were all correlated with immune signaling pathways. We have accepted the suggestion and evaluated the expression of these immune-associated lncRNA biomarkers in single-cell sequencing data (*Diether et al., 2018, Nature Medicine*). First, we calculated the average expression of these lncRNAs in B cells or T cells. Next, the same number of lncRNAs were randomly selected, and we calculated their average expression. This process was repeated 100,000 times. We found that the lncRNAs used for classification showed significantly higher average expression than randomly selected lncRNAs in B cells and T cells (Supplementary Fig. 18). These results suggest that these lncRNAs are likely to be associated with immunology, further validating the ImmLnc pipeline. We have discussed this issue in our revised manuscript.

Supplementary Fig. 18. The average expression of lncRNA biomarkers was significantly higher than that of randomly selected lncRNAs in B cells and T cells. The lines show the distribution of average expression in random conditions. The red dots represent the observed average expression levels.

15. Fig. 5E - The authors claim that patients in the C1 group have significantly poorer prognosis than other patients. However, the difference is too small to claim the significance.

Reply: Many thanks for this suggestion. As the reviewer commented, the survival difference is small, although it is significant ($P = 0.02$). Based on the suggestion from another reviewer, we discussed this in-depth. When we compared the survival difference between C1 and C2 subtypes, we found that the C2 patients showed better survival than C1 patients (log-rank $P = 0.009$). In addition, we found that there were approximately twice as many C2 patients as C1 patients alive after 4000 days. When we considered the chemotherapy patients, we found that the C1 patients receiving chemotherapy had significantly poorer prognosis (log-rank $P = 0.0002$). We have discussed these results in our revised manuscript.

16. Fig. 6B - Authors compared the HRD scores across different subtypes and found that patients in the C2 subtype were with significantly higher HDR scores. However, this bar chart does not support their claim. There were very low/no differences in HDR scores between C1 and C2 groups.

Reply: Thanks for this suggestion. We found that C1 and C2 patients had significantly higher HDR scores than C3 patients. We have accepted the suggestion and made our claim clearer in our revised manuscript.

17. In Fig 1, the title of section 2, “lncRNA Regulators are Likely to Expression Perturbation in Cancer” is grammatically incorrect.

Reply: Many thanks for this suggestion. The manuscript has been reviewed and proofread by several professionals, and we have mad use of the editorial services of Accdon/LetPub to avoid any potential academic or grammatical errors.

Reply to Reviewer #2:

Overall summary

In this manuscript, Li et al develop a pipeline “ImmLnc” to identify lncRNAs that may regulate the immune response in cancer, by identifying lncRNAs that correlate with specific immune gene expression signatures and immune cell infiltration. The authors used this approach to examine non-small cell lung cancer more closely, and were able to identify three subgroups of lung cancer with different intrinsic features (e.g. mutational burden) and differing immune responses.

Overall the approach taken in this manuscript appears to be largely sound and may be a useful methodology to apply when investigating lncRNAs in cancer, with clearly presented figures and most data/methods described sufficiently. However, the manuscript contains a large number of grammar errors that make it difficult to understand results/methodology at certain points, and would benefit from a substantial rewriting of the text. Furthermore, there are several experimental/analytical points that should be addressed before publication.

Reply: We sincerely appreciate the encouraging and constructive comments. We address the comments point-by-point below.

Major Points

1. *The manuscript would greatly benefit from further proofreading and/or rewriting to improve the English throughout.*

Reply: Many thanks for this suggestion. The manuscript has been reviewed and proofread by several professionals, and we have made use of the editorial services of Accdon/LetPub to avoid any potential academic or grammatical errors.

2. *One key question not addressed is whether the lncRNAs identified as correlating with immune responses are derived from host cells or infiltrating immune cells. Indeed, there is no data presented anywhere in the manuscript derived from pure immune cell populations. Given that there are published RNA-seq datasets available for the majority of the immune cells identified as tumour-infiltrating cells, these could be examined to identify for example which lncRNAs correlating with CD8 T cell infiltration are specifically expressed in CD8 T cells and not the host tumour.*

Reply: Thanks for this suggestion. We have accepted the suggestion and investigated the expression of lncRNAs in immune cell populations, which were identified by single-cell sequencing (*Diether et al., 2018, Nature Medicine*). We found that the lncRNAs that were positively correlated with B cell or T cell infiltration were significantly more highly expressed in B cell or T cell populations (Supplementary Fig. 10). In addition, we derived the lncRNA expression profile in immune cells of TCGA samples based on the ideas from RESPECTEx (*Ng et al., 2019, Nucleic Acids Research*). We found that the immune cell infiltration-related lncRNAs tend to be more highly expressed in

immune cells than other lncRNAs (Supplementary Fig. 9). These results further suggest that the identified lncRNAs are likely to be highly expressed in immune cells. We have provided these results in our revised manuscript.

Supplementary Fig. 9. Heat map showing the ratio between average expression of immune cell infiltration-related and other lncRNAs.

Supplementary Fig. 10. The expression of lncRNAs in immune cell populations. a. The expression of lncRNAs that were correlated with immune cell infiltration in LUSC. b. The expression of lncRNAs that were correlated with immune cell infiltration in LUAD. *** $P < 0.001$, Wilcoxon rank sum test.

3. The authors take a very broad correlative approach in this study, however some consideration should be given to the fact that most lncRNAs are expressed at very low levels. This is particularly pertinent when considering (following on from point 2) that many of the identified lncRNAs may be derived from tumour-infiltrating cells that may make up a relatively small fraction of the overall tumour. Can the authors provide some further evidence that a proportion of the lncRNAs at least can be detected at sufficiently robust expression levels to act as biomarkers?

Reply: Thank you very much for this suggestion. We have accepted the suggestion and investigated the expression range of immune-related lncRNAs across cancer types (Supplementary Fig. 24a). We found that immune-related lncRNAs are likely to be expressed at higher levels across cancer types. Particularly, when we divided lncRNAs into two groups (high vs low), we found that immune-related lncRNAs are significantly enriched in the groups with higher expression (Supplementary Fig. 24b, $P < 0.001$). In addition, we explored the expression levels of the 28 lncRNAs identified in nonsmall cell lung cancer. We found that these lncRNAs had $\log_2(\text{FPKM}+0.05) > 5$ in $>75\%$ of the patients (Supplementary Fig. 24c). These results suggest that a

proportion of lncRNAs can be detected at sufficiently robust expression levels to act as biomarkers. We have discussed this issue in the Discussion section of our revised manuscript.

Supplementary Fig. 24. Immune-related lncRNAs exhibit high expression in cancer. a. The proportion of lncRNAs with different expression levels in cancer. The left panel shows all lncRNAs, and the right panel shows immune-related lncRNAs. b. The odds ratio for Fisher's exact test. c. The cumulative distribution of the expression of 28 lncRNAs.

4. lncRNAs are broadly grouped together throughout the manuscript with no subclass analysis made, apart from after line 193 when an unexplained switch to intergenic lncRNAs is made. One might expect that certain subclasses of lncRNAs (e.g. enhancer RNAs, promoter-associated lncRNAs) to naturally correlate more with the expression of proximal genes in the immune response. The authors should try and address whether there are subclasses of lncRNAs that are more likely to be correlative than others.

Reply: Thank you very much for this comment. We have accepted the suggestion and calculated the proportion of subclasses of lncRNAs in immune-associated lncRNAs. We found that more than 75% of the immune-associated lncRNAs were antisense and/or intergenic (Supplementary Fig. 14). The antisense lncRNAs overlap with coding genes and might play similar functions as coding genes. Thus, we focused on intergenic lncRNAs in our analysis. In addition, we also reanalyzed the results based on all lncRNAs. We found that the cancer-related lncRNAs have

significantly higher rank than other lncRNAs (Supplementary Fig. 16). We have discussed this in our revised manuscript.

Supplementary Fig. 14. The proportion of lncRNAs in different subtypes. The lncRNA classification information was obtained from GENCODE.

Supplementary Fig. 16. The relative ranks of lncRNAs based on the number of cancer types that show an association with immune pathways.

5. Some of the conclusions in the last section of the manuscript (figures 5 and 6) seem somewhat over-interpreted in places. Some more specific points are raised below, but whilst the C1 cluster appears immunologically distinct from the rest of the cases, I am not convinced that there is any real differences

between the C2 and C3 classes.

Reply: Many thanks for this comment. Our inappropriate and unclear description might confuse the readers. As the reviewer commented, we found C1 patients are immunologically distinct from C2 patients and exhibit higher proliferation features. C2 patients exhibit more immune-related features, such as higher mutation burden and immune, CYT, and MHC scores. These results suggest that C2 is likely to be the immune subtype. Although C3 patients exhibit features similar to those of C2 patients, we found some differences between C2 and C3 subtypes, such as mutation burden and HRD score. Therefore, we defined C3 as the intermediate subtype. To make this clear, we have revised the sentence and discussed the difference in-depth to avoid misinterpretation in the revised manuscript.

6. Although the methods generally contain sufficient detail, some more explanation of key analyses should be made briefly in certain places in the manuscript and/or figure legends to aid the readers with understanding the points being made.

Reply: Many thanks for this suggestion. We have accepted the suggestion and provided sufficient details in the Methods and Figure Legends sections of our revised manuscript.

Minor points

The following points are broken down by lines in the manuscript:

44 – This is a somewhat contentious statement that derives from early studies into the lncRNA field, now widely debunked (see Graur et al, 2013, Genome Biol Evol for example) . It may be truer if aggregating all possible cell types, but in a given cell type at any one time the majority of the genome will remain untranscribed. I would remove or soften this statement.

Reply: Thank you very much for this suggestion. We have accepted the suggestion and removed this statement in our revised manuscript.

86- define/describe RS score in manuscript. Some further description of the correlation scores and GSEA analysis would be helpful too as it's a key point in the analysis.

Reply: Many thanks for this suggestion. We have accepted the suggestion and provided a detailed definition of RS score and GSEA analysis in our revised manuscript.

125-128 – it would be good to know the no. of lncRNA-pathways relationships that did not overlap between matched datasets (i.e. provide percentage figures).

Reply: Thank you very much for pointing this out. We have accepted the suggestion and provided the percentage figures and the data used for all figures in our revised manuscript.

Figure 2. Validation of immune-related lncRNAs. b. The overlap of immune–lncRNA pathways in independent datasets of the same cancer type. The top bar plots show the observed/expected values of the hypergeometric test. The bottom Venn plots show the number of overlapping lncRNA–pathway pairs. *** $P < 0.01$.

139 – *In supplementary figure 2d, DLBC and LAML are highlighted as similar – I'd question whether these are really similar at all – these are very different cancers.*

Reply: Thanks for this suggestion. As the reviewer commented, the co-existence of diffuse large B cell lymphoma (DLBC) and acute myeloid leukemia (LAML) is extremely rare. However, several recent studies have reported several cases of concurrence of these two cancers (*Khadega et al., J Med Case Rep, 2018; Junichi et al., Intern Med, 2018*). Here, we found that the immune-related lncRNAs of these two cancer types were significantly overlapped. Such molecular events can be utilized as surrogate biomarkers for early detection. We have discussed this in the revised manuscript.

148 – *It would be good to see the identification of immune infiltrating subsets validated by an orthogonal approach e.g. CIBERSORT.*

Reply: Many thanks for this suggestion. We have accepted the suggestion and identified the lncRNAs that were correlated with immune cell infiltration based on CIBERSORT. We found significant overlap between two lncRNA sets (Supplementary Table 5). We have discussed this in our revised manuscript.

162-164 – *I do not understand the point being made in this sentence.*

Reply: Thank you very much for this comment. Our inappropriate and unclear description might confuse the readers. We compared the proportion of lncRNAs that were correlated with immune cell infiltration in immune-related and other lncRNAs by Fisher's exact test. We found that the proportion is higher in immune-related lncRNAs. To make this result clearer, we have rewritten this sentence in our revised manuscript.

193 – *Where do these 5050 lincRNAs derive from? Why the switch to intergenic lncRNAs only?*

Reply: Thanks for this suggestion. The annotation and classification of lncRNAs were downloaded from GENCODE. We found that more than 75% of the immune-associated lncRNAs were antisense and/or intergenic (Supplementary Fig. 14). The antisense lncRNAs overlap with coding genes and may exhibit similar functions as coding genes. Therefore, we focused on intergenic lncRNAs in our analysis. In addition, we also reanalyzed the results based on all lncRNAs and obtained the similar results (Supplementary Fig. 16). These results are provided in our revised manuscript.

Supplementary Fig. 14. The proportion of lncRNAs in different subtypes. The lncRNA classification information was obtained from GENCODE.

Supplementary Fig. 16. The relative ranks of lncRNAs based on the number of cancer types that show an association with immune pathways.

238-239 – how were the tumour differentiation, hippo pathway, stem and cell cycle activity scores calculated? I cannot see these described in the methodology

Reply: Thanks for this suggestion. The differentiation, hippo pathway activity, stemness, and cell cycle activity scores were obtained from the supplemental tables of one recent study (*Che et al., 2017, Oncogene*). We have provided the methods and references in our revised manuscript.

242 -244 – the differences in survival between groups appears relatively modest, however this may be exacerbated by the extreme length of the follow up and its impact on the curve. It might be worth considering how many patients are actually alive after say 3000 or 4000 days, and whether any events after this point are relevant to the cancer diagnoses (i.e. by the point patients may be succumbing to old age).

Reply: Thanks for this suggestion. We have accepted the suggestion and discussed this in-depth. When we compared the survival difference between C1 and C2 subtypes, we found that the C2 patients were with better survival than C1 patients (log-rank $P = 0.009$). In addition, we found that there were approximately twice as many C2 than C1 patients alive after 4000 days. These results are discussed in our revised manuscript.

259 – HRD scores for C2 higher than C3 but not C1 as implied

Reply: Many thanks for this suggestion. As the reviewer commented, we found the HDR scores was significantly higher in C1 and C2 patients than C3 patients. We have rewritten this sentence to make this result clearer in our revised manuscript.

261-263 – where are these 160 signatures derived from?

Reply: Many thanks for this suggestion. These signatures were obtained from the TCGA pan-cancer study (*Thorsson et al., 2018, Immunity*). We have provided the references in our revised manuscript.

265-269. What about C3 group? These are also generally high in B cell signatures at least and should be discussed.

Reply: Thanks for this suggestion. We have accepted the suggestion and discussed this in-depth for the C3 group in our revised manuscript.

277-278 – Differences in response are very modest, particularly in comparison to C3 group. Can the authors provide a kaplan meier curve analysis instead?

Reply: Many thanks for this suggestion, we have accepted the suggestion and compared the survival for chemotherapy patients of different subtypes. We found a significant difference in survival rate (log-rank $P = 0.0002$). We have discussed this in our revised manuscript.

323-326 – having new data appear in the discussion is confusing and should be mentioned earlier in the results section or be removed.

Reply: Many thanks for this suggestion. We have accepted the suggestion

and removed this result from the Discussion section in our revised manuscript.

416-421- The description of identifying differentially expressed lncRNAs is confusing and does not sound stringent e.g. defining a lncRNA as being expressed if it is >0 is a very low threshold.

Reply: Thank you very much for this comment. Our inappropriate and unclear description might confuse the readers. We filtered the lncRNAs with FPKM = 0 in >30% of the samples in our analysis. In addition, we also used FPKM = 1 as a threshold and obtained similar results.

726-728 – Figure 3 legend – it is not clear to me what is being shown in the inset bar-graph – please reword description

Reply: Thanks for this suggestion. We have provided the details in the figure legends in our revised manuscript.

744 – 746 – Figure 5 legend – please provide more detail on the calculated scores.

Reply: Thank you very much for this suggestion. We have provided the details in the Methods section of our revised manuscript.

REVIEWERS' COMMENTS:

Reviewer #1 (Remarks to the Author):

The authors have satisfactorily addressed all of my comments and concerns.

Reviewer #2 (Remarks to the Author):

In this revision, Li et al do appear to have addressed the majority of my concerns, bar the following few comments:

Line 50 – As stated in previous review, even though tens of thousands of different RNA transcripts may be transcribed in an individual cell, the majority of the genome is not transcribed. Please reword or remove

Lines 156-161 – This section is incorrect and should be removed. The original point and statement was that AML and DLBCL are closely related tumour types. This is true in so far that they are both derived from immune cells. However, AML is a disease originating from myeloid precursor cells, which populates the bone marrow and circulation, and DLBCL is a disease originating from mature B cell lymphocytes that is largely confined to tumours within lymph nodes. The authors have referenced two case studies on co-occurrence of leukaemia and DLBCL. These are however exceptional cases and far from the norm. Furthermore, the second paper referenced refers to Chronic Myeloid Leukaemia and Chronic Lymphocytic Leukaemia, and not AML. There is no evidence that these tumours, in the vast majority of patients, are derived from the same cell of origin. The similarity of these tumour types is of minor consequence to the conclusions of the manuscript, and so it would be much better if any reference to these was removed.

Supplementary Figure 7 - I completely misunderstood this graph until I looked up the referenced publication, which focused on deconvoluting immune cell gene expression from bulk RNA-seq data. The methodology used should be better described, as it can be easily misinterpreted at the moment

Line 254, 398 and 399 – the authors use the phrase “regulates” in several places, whereas it would be more appropriate to say that the identified lncRNAs “correlate” or “associate with” immune pathways

Grammatical errors:

Line 31 – “prioritizing” should be “prioritize”, and “identify” should be “identified”.

Line 41 – Remove “It is found that”, and start sentence with “Gene expression”

Line 57- “a recent study has reported that the tumor microenvironment play important roles in cancer development”. This sentence sounds like there has only been a single study in this field, whereas in reality there have been hundreds, if not thousands, of studies into the immune microenvironment. Please reword appropriately

Line 121 – “identified in more cancer types”. This doesn’t entirely make sense, should probably say “multiple” instead.

Line 142 – change “significantly” to “significant”

Line 188 – change “was” to “were”

Line 196 – change “immune-related lncRNAs is” to “immune-related lncRNAs are”

Reply to Reviewer #1 (Remarks to the Author):

The authors have satisfactorily addressed all of my comments and concerns.

Reply: Thank you very much for your efforts on our manuscript.

Reviewer #2 (Remarks to the Author):

In this revision, Li et al do appear to have addressed the majority of my concerns, bar the following few comments:

Reply: Thank you very much for your valuable comments. The manuscript has been greatly improved by addressing the reviewers' comments.

Line 50 – As stated in previous review, even though tens of thousands of different RNA transcripts may be transcribed in an individual cell, the majority of the genome is not transcribed. Please reword or remove

Reply: Thanks for this suggestion. We have accepted the suggestion and removed this in our revised manuscript.

Lines 156-161 – This section is incorrect and should be removed. The original point and statement was that AML and DLBCL are closely related tumour types. This is true in so far that they are both derived from immune cells. However, AML is a disease originating from myeloid precursor cells, which populates the bone marrow and circulation, and DLBCL is a disease originating from mature B cell lymphocytes that is largely confined to tumours within lymph nodes. The authors have referenced two case studies on co-occurrence of leukaemia and DLBCL. These are however exceptional cases and far from the norm. Furthermore, the second paper referenced refers to Chronic Myeloid Leukaemia and Chronic Lymphocytic Leukaemia, and not AML. There is no evidence that these tumours, in the vast majority of patients, are derived from the same cell of origin. The similarity of these tumour types is of minor consequence to the conclusions of the manuscript, and so it would be much better if any reference to these was removed.

Reply: Thanks for this suggestion. We have accepted the suggestion and removed this conclusion from our revised manuscript.

Supplementary Figure 7 - I completely misunderstood this graph until I looked up the referenced publication, which focused on deconvoluting immune cell gene expression from bulk RNA-seq data. The methodology used should be better described, as it can be easily misinterpreted at the moment.

Reply: Thanks for this comment. We have provided more details about this method in our revised manuscript in supplementary methods.

Page 7

Moreover, we derived the lncRNA expression profile in immune cells of TCGA bulk RNA-Seq samples based on the ideas from RESPECTEx²⁷ (see details in Supplementary methods). We also found that these lncRNAs were significantly more highly expressed in immune cells (Supplementary Fig. 7).

Line 254, 398 and 399 – the authors use the phrase “regulates” in several places, whereas it would be more appropriate to say that the identified lncRNAs “correlate” or “associate with” immune pathways

Reply: Thank you very much for this suggestion. We have accepted the suggestion and replaced “regulates” by “associate with” in our revised manuscript.

Page 9

Based on the ImmLnc method, we revealed that it is associated with the TCR signaling pathway in 16

cancer types (Fig. 4e).

Page 14

First, the predicted results indicate that not all immune-related pathways are equally associated with lncRNAs; cytokine and cytokine receptor pathways are likely to be correlated with more lncRNAs.

Grammatical errors:

Line 31 – “prioritizing” should be “prioritize”, and “identify” should be “identified”.

Line 41 – Remove “It is found that”, and start sentence with “Gene expression”

Line 57- “a recent study has reported that the tumor microenvironment play important roles in cancer development”. This sentence sounds like there has only been a single study in this field, whereas in reality there have been hundreds, if not thousands, of studies into the immune microenvironment. Please reword appropriately

Line 121 – “identified in more cancer types”. This doesn’t entirely make sense, should probably say “multiple” instead.

Line 142 – change “significantly” to “significant”

Line 188 – change “was” to “were”

Line 196 – change “immune-related lncRNAs is” to “immune-related lncRNAs are”

Reply: Thank you very much for these suggestions. We have accepted the suggestion and revised these issues in our revised manuscript. The manuscript has been reviewed and proofread by several professionals to avoid any potential academic or grammatical errors.